# Linking air stagnation in Europe with the synoptic- to large-scale atmospheric circulation

Jacob W. Maddison[1], Marta Abalos[1], David Barriopedro[2], Ricardo García-Herrera[1,2], Jose M. Garrido-Perez[1], and Carlos Ordóñez[1]

[1]Department of Earth Physics and Astrophysics, Universidad Complutense de Madrid, Madrid, Spain
[2]Instituto de Geociencias (IGEO), CSIC-UCM, Madrid, Spain
**Correspondence:** Jacob Maddison (JACOBMAD@ucm.es)

**Abstract.** The build-up of pollutants to harmful levels can occur when meteorological conditions favour their production or accumulation near the surface. Such conditions can arise when a region experiences air stagnation. The link between European air stagnation, air pollution, and the synoptic- to large-scale circulation is investigated in this article across all seasons and the 1979–2018 period. Dynamical based indices identifying atmospheric blocking, Rossby wave breaking, subtropical ridges, and the North Atlantic eddy-driven and subtropical jets are used to describe the synoptic- to large-scale circulation as predictors in statistical models of air stagnation and pollutant variability. It is found that the large-scale circulation can explain approximately 60% of the variance in monthly air stagnation, ozone and wintertime particulate matter (PM) in five distinct regions within Europe. The variance explained by the model does not vary strongly across regions and seasons, apart from for PM when the skill is highest in winter. However, the dynamical indices most related to air stagnation do depend on region and season. The blocking and Rossby wave breaking predictors tend to be the most important for describing air stagnation and pollutant variability in northern regions whereas ridges and the subtropical jet are more important to the south. The demonstrated correspondence between air stagnation, pollution and the large-scale circulation can be used to assess the representation of stagnation in climate models, which is key for understanding how air stagnation and its associated climatic impacts may change in the future.

## 1   Introduction

Poor air quality poses one of the largest environmental threats to public health. Long term exposure to air pollutants such as particulate matter (PM) and ozone can cause severe cardiovascular and respiratory diseases and is responsible for between two and four hundred thousand premature deaths every year in Europe (World Health Organization, 2011; Giannadaki et al., 2016; European Environment Agency, 2020). Risks depend strongly on the weather, increasing when meteorological conditions favour the production or accumulation of the pollutants in the lowest part of the atmosphere (Ordóñez et al., 2005; Jacob and Winner, 2009; Weaver et al., 2009; Barmpadimos et al., 2011; Dawson et al., 2014). Such conditions can arise when a stable air mass becomes settled over a region and remains quasi-stationary for an extended amount of time, often referred to as *air stagnation*. It is therefore key to characterize air stagnation, understand the dynamics driving its development and quan-

tify its importance for pollution events. Furthermore, climate change is expected to increase the occurrence of air stagnation
(Mickley et al., 2004; Leung and Gustafson Jr, 2005; Horton et al., 2012, 2014; Caserini et al., 2017; Gao et al., 2020; Lee
et al., 2020), emphasising the need to fully understand its occurrence and variability. Previous studies have mainly focused on
local conditions when describing air stagnation, without considering synoptic-scale structures and large-scale features of the
atmospheric circulation. In this article, we use the synoptic- to large-scale atmospheric circulation to describe the variability of
air stagnation within distinct regions of Europe.

Globally, regions in the tropics generally experience the highest air stagnation frequencies (Horton et al., 2012, 2014), though
several regions in the midlatitudes, such as North America, China and the Mediterranean, have stagnation frequencies over 40%
(e.g., Horton et al., 2012, 2014; Huang et al., 2017; Garrido-Perez et al., 2018). Within Europe, stagnation exhibits distinct
spatial and temporal characteristics. Regions in the south experience frequent stagnation whilst the frequency is generally
lower in northern regions (Horton et al., 2012, 2014; Garrido-Perez et al., 2018). Describing the conditions necessary for air
stagnation is relatively straight forward, but objectively identifying them in data can be more complicated, as it usually relies
on indices based on predetermined, arbitrary thresholds of meteorological variables that characterise stagnation (e.g. Wang and
Angell, 1999; Horton et al., 2012, 2014; Wang et al., 2016, 2018; Huang et al., 2017). Commonly used air stagnation indices
(ASIs) include terms based on the wind speed at 10 m (Wang and Angell, 1999; Horton et al., 2012; Wang et al., 2016, 2018),
wind speed in the boundary layer (Huang et al., 2018), boundary layer height (Wang et al., 2016, 2018) and precipitation (Wang
and Angell, 1999; Horton et al., 2012; Wang et al., 2016, 2018; Huang et al., 2018). However, similar local weather conditions
can occur under very different rearrangements of the large-scale flow. Therefore, there is a need to bridge the gap between local
air stagnation in Europe and the large-scale atmospheric circulation.

The extent to which air stagnation is related to air pollution has recently come under scrutiny in the literature. Intuitively, the
weak winds in the lower troposphere and the absence of precipitation during air stagnation should provide ideal conditions for
pollutants to accumulate near the surface. Indeed, many previous studies have demonstrated a relationship between stagnation
and pollution events across the globe. For Europe, stagnation has been shown to cause clear increases in both winter $PM_{10}$ and
summer ozone, by between 31–63% and 12–23% respectively, dependent on the region (Garrido-Perez et al., 2018, 2021). A
close relationship exists between stagnation and ozone on interannual timescales for most of Europe (Garrido-Perez et al., 2018)
and on daily timescales for central and southern regions (Garrido-Perez et al., 2019) (with correlations between 0.5 and 0.8).
Annual variability in both $PM_{2.5}$ and ozone were shown to be strongly related to the variability in air stagnation (correlations
of 0.68 and 0.79, respectively) over Eastern North America in Schnell and Prather (2017). Furthermore, the persistence of air
stagnation can have a large impact on ozone in this region, with levels of the pollutant increasing with each stagnant day (Sun
et al., 2017). Air stagnation has also been demonstrated to be important for the build up of pollutants in cities in China (Huang
et al., 2018; Liao et al., 2018; Wang et al., 2018), India (Kanawade et al., 2020) and Chile (Toro et al., 2019).

However, some studies find a weaker relationship between air stagnation and pollutant levels. Stagnation was not identified
as a strong predictor in a statistical model of summer ozone levels in the Northeastern United States (US) in Oswald et al.
(2015), with temperature and solar radiation better predictors. In agreement with this result, Kerr and Waugh (2018) showed
that the temporal correlations between air stagnation and both $PM_{2.5}$ and ozone in the US are quite weak, whilst positive

(ranging from around 0.1 to 0.6). The effect of stagnation on pollution can also depend on season, with $PM_{10}$ anomalies during stagnation in Europe lower in summer than in winter (Garrido-Perez et al., 2021), and region, with stagnation being a better predictor of summer ozone in central and southern Europe than in northern Europe (Garrido-Perez et al., 2019). Some of these studies are limited by the use of only one index to identify air stagnation. The degree to which air stagnation impacts air pollution has been shown to be sensitive to the choice of ASI (Huang et al., 2018; Wang et al., 2018; Garrido-Perez et al., 2021). Garrido-Perez et al. (2021) found differences in the seasonal cycle and effect on pollutant build up for three ASIs in Europe, suggesting results obtained using a single ASI should be taken with some caution. In light of this, the analyses included in this article begin by considering stagnation as defined in the Horton et al. (2012) ASI, as this index is known to relate to pollutant levels in Europe. The results are then compared for stagnation as defined in two additional ASIs, and for direct estimates of concentrations of the pollutants $PM_{2.5}$ and ozone. This ensures that we can identify features of the synoptic- to large-scale flow that are truly important for air quality and not a feature of a particular ASI, as well as contributing to the debate on the usefulness of air stagnation as a proxy for air pollution events.

The large-scale flow is inherently related to air stagnation occurrence in Europe and thus also to air quality. Their relation can be summarised by considering the dynamics of midlatitude weather. The North Atlantic jet streams and Rossby waves are the main drivers of midlatitude weather, and are associated with high-impact weather systems such as atmospheric blocking and extratropical cyclones, which exert a strong control over surface conditions (Hoskins et al., 1985). Blocking events typically occur when a large-scale ridge in an upper-level Rossby wave develops (e.g. Woollings et al., 2018). They are an obvious candidate for driving air stagnation events as they are characterised by a synoptic-scale, quasi-stationary anticyclone (Rex, 1950), and thus provide the weak winds and absence of precipitation that define air stagnation. Blocks have been shown to increase pollutant levels in Europe (Hamburger et al., 2011; Garrido-Perez et al., 2017; Ordoñez et al., 2017; Webber et al., 2017; Vautard et al., 2018), the United States (Comrie and Yarnal, 1992), and in Asia (Yun and Yoo, 2019). The position of the North Atlantic jet stream (Ordóñez et al., 2019), the presence of subtropical ridges (Garrido-Perez et al., 2017; Ordoñez et al., 2017), and the passage of midlatitude cyclones (Leibensperger et al., 2008; Tai et al., 2010, 2012; Leung et al., 2018) can also influence air stagnation development and pollution levels.

With this in mind, the aims of this study are (i) to complete a comprehensive identification and comparison of the synoptic- to large-scale drivers of air stagnation within Europe, (ii) to quantify the amount of monthly variability in air stagnation that can be explained by the large-scale circulation, (iii) to assess the sensitivity of the results to the use of air stagnation index, and (iv) to test the robustness and potential implications of the results by replacing the ASI (a pollution proxy) with direct pollutant data.

The article is presented as follows. Section 2 contains a description of the data, the various circulation indices used throughout the article and the statistical model used. The large-scale circulation patterns identified during air stagnation are discussed in Section 3. In Section 4, the statistical model is analysed and its results explained in detail for a specific case. The results obtained for models of two additional ASIs and two pollutants are compared and discussed in Section 5. The article is concluded in Section 6.

## 2 Data and methods

Several indices are used in this study to identify both air stagnation and relevant dynamical features. A brief description of each index is included but the reader is referred to the referenced papers for further information. The majority of data used in this study is from the European Centre for Medium Range Weather Forecasts' (ECMWF) ERA5 reanalysis (Hersbach et al., 2020). This data is interpolated onto a one degree grid for the analysis and covers the period 1979–2018, and the North Atlantic and European region ($100°$W – $80°$E, 20–85° N). Pollutant data analysed in this study is taken from the ECMWF global reanalysis of atmospheric composition, the Copernicus Atmosphere Monitoring Service (CAMS) reanalysis (Flemming et al., 2017), and covers the period 2003–2018. Daily mean mass concentrations of $PM_{2.5}$ and daily mass mixing ratios of ozone at 1500 UTC (when levels typically peak) are downloaded for the analyses, as well as the monthly averages of daily mean $PM_{2.5}$ and 1500 UTC ozone. Ozone data are converted to volume mixing ratios before the analyses presented here.

### 2.1 Air stagnation indices

The ASI introduced in Horton et al. (2012) is used for most of the analyses presented in this study. A grid point is defined as stagnant if: (i) the daily mean wind speed at 10 m is less than 3.2 m/s, (ii) the daily mean wind speed at 500 hPa is less than 13.0 m/s, and (iii) total daily precipitation is less than 1 mm (a dry day). Values at 0000, 0600, 1200 and 1800 UTC are averaged to calculate the daily mean wind speeds and hourly precipitation totals are summed to calculate the daily precipitation. We compare our findings with two other ASIs to ensure the robustness of their results.

The ASI by Wang et al. (2016, 2018) defines stagnation when precipitation is below 1 mm and the boundary layer height is below a certain threshold. The threshold is a function of season and wind speed at 10 m (see Wang et al. (2018) for further details). Huang et al. (2018) use an ASI that again requires daily precipitation less than 1 mm. For stagnation, it is also required that the vertical integral of the horizontal wind speed within the boundary layer is less than 6000 $m^2$/s, and that there is no potential thunderstorm activity on that day. Thunderstorm activity is ruled out by excluding days with CAPE > 100 J/kg and CIN > − 50 J/kg. These values have been modified for European stagnation following Taszarek et al. (2018) and Garrido-Perez et al. (2021).

### 2.2 Dynamical indices

To link the large-scale circulation to air stagnation we use several daily dynamical indices that describe key features of the extratropical flow. Blocking, Rossby wave breaking (RWB), subtropical ridges, and the latitude and speed of the eddy-driven and subtropical North Atlantic jets are identified using indices based on various meteorological variables. These features of the large-scale circulation do not represent the full spectrum of synoptic systems that characterize the midlatitude atmospheric circulation, but are chosen because of their known association with air stagnation (Section 1).

*Atmospheric blocking* is identified using the index of Scherrer et al. (2006). A block is identified using instantaneous meridional gradients in geopotential height at 500 hPa ($Z500$) (at 1200 UTC). The method looks for the overturning of the geopotential height contours in the midlatitudes (between 35 and 75°N) characteristic of a block and defines a grid point as blocked

when the gradient from the south is positive and the gradient to the north is strongly negative (less than -10 m/degree). A persistence criteria of 3 days is used to identify block events.

    *RWB* is calculated following the definition of Masato et al. (2012). Using the daily average potential temperature field on the dynamical tropopause (potential vorticity surface at 2 PVU, $\theta_{2PVU}$), RWB is identified when the meridional gradient of potential temperature is reversed. Although RWB often occurs in association with blocking, these events are typically

persistent. By not imposing persistence criteria, we also account for transient RWB events, including those associated with cyclone development (e.g. Gómara et al., 2014). In addition, RWB captures persistent flow reversals dominated by cyclonic wave breaking that may be missed by the blocking index, which is known to be biased towards anticyclonic blocks and can miss some omega type blocks (Barriopedro et al., 2010).

    *Subtropical ridges* are classified following a slightly modified method of that developed by Sousa et al. (2018). Ridges are

defined as positive $Z500$ anomalies in the subtropical midlatitudes (south of 50°N in all seasons except summer when 55°N is used) that do not extend further poleward. The local 60th percentile of the daily $Z500$ series (smoothed with a 31-day running mean) is employed to identify positive anomalies. A ridge is then identified when more than half of the subtropical midlatitude grid points but less than half of the northern midlatitude grid points are above their 60th percentile in three separate longitudinal sectors (Atlantic, ATL, 30°W–0°, European, EUR, 0°–30°E, and Russian, RUS, 30°E–60°E). These criteria are

chosen to avoid double counting blocks (i.e. to ensure blocks are not detected as subtropical ridges). The less strict thresholds for identifying ridges compared to those in the original definition of Sousa et al. (2018) were chosen to increase the frequency of ridges, therefore providing a more balanced frequency of events (as compared with blocking and RWB), which in turn is expected to yield more robust linkages in the statistical model. Defining a subtropical ridge using two or one longitudinal sectors (spanning the same longitudes) does not have a large impact on the results presented here.

The *eddy-driven jet speed* and *latitude* are calculated using the index introduced in Woollings et al. (2010). The daily mean zonal wind is averaged vertically (between 925 and 700 hPa with data every 75 hPa) and zonally (between 0 and 60°W). A 10-day smoothing is then applied to remove the influence of individual synoptic systems. The magnitude and latitude of the maximum wind speed of the resulting meridional profile are selected as the eddy-driven jet speed and latitude, respectively.

    The *subtropical jet speed* and *latitude* are described using the zonal wind at 250 and 200 hPa. Following the method used

to describe the eddy-driven jet, the daily mean zonal wind is averaged vertically (between 250 and 200 hPa) and zonally (between 20°E and 60°W, the longitudinal range where the subtropical jet is typically identified (e.g. Asiri et al., 2020)) and then low-pass filtered. . The magnitude and latitude of the maximum wind speed of the resulting meridional profile are selected as the subtropical jet speed and latitude, respectively. The eddy-driven and subtropical jet can at times be indistinguishable, particularly in summer, but in many cases they are expected to be separated (as inferred from the climatological means: see

also Molnos et al. (e.g. 2017)). Furthermore, the eddy-driven jet speed and upper-level wind speed will likely more strongly influence the lower- and upper-level wind criteria of the ASI, respectively.

    Together, these dynamical indices provide a comprehensive picture of the large-scale circulation over Europe. We construct a multiple linear regression (MLR) model with these indices as predictors of the variability in air stagnation and air pollution. Blocks, RWB and ridges are referred to as regional predictors as they are identified within each region, whilst the eddy-driven

and subtropical jet are called large-scale (or Europe-wide) as they are defined in areas covering more than one of the regions. Using all of the predictors ensures we include major factors that are favourable for air stagnation but means that they aren't necessarily independent. For example, the blocking and Rossby wave breaking indices or the eddy-driven and subtropical jet indices may at certain times identify the same large-scale feature. Their collinearity and relative importance for stagnation will be taken into account by the MLR model, as described in the next section.

## 2.3 Stepwise multiple linear regression

MLR models are used to study the linear relationship between a chosen response variable and a set of predictor variables. Here we use the monthly series of air stagnation or pollutant concentration as the response variable and the set of dynamical indices as predictor variables. The MLR model thus takes the form

$$Y = \beta_0 + \beta_1 I_1 + \beta_2 I_2 + ... + \beta_k I_k, \tag{1}$$

where $Y$ is the air stagnation or pollutant time series, each $I$ represents a dynamical index, $\beta_0$ is the model intercept and the remaining $\beta$ terms are the regression coefficients. The regression coefficients are estimated using a least-squares approach (Montgomery et al., 2012). This regression analysis is applied to model the monthly variability of stagnation and air pollution for separate regions and seasons, as described in Sections 4 and 5.

A stepwise approach has been used in Section 4 to select the dynamical indices that account for the largest variance of air stagnation from the total set of predictors. Stepwise approaches have been used to model the variability of pollutants in Europe (Barmpadimos et al., 2011, 2012; Otero et al., 2016; Garrido-Perez et al., 2021). The goal of the stepwise approach is to inform which of the predictors can be excluded from the MLR model without losing a significant amount of the variance explained by the model. The method consists of five steps. **(i)** A linear model for air stagnation is constructed for each of the dynamical indices separately and the index providing the most skill (highest $R^2$) is selected. **(ii)** The remaining indices are added one at a time to construct MLR models with an additional predictor. The added index that yields the most skillful model is selected. **(iii)** The choice from step (i) is verified. We remove the first variable from the model and construct models with the variable selected from step (ii) and each of the remaining variables separately. If any of the new models have a higher $R^2$ than that constructed in step (ii), the new variables replace those previously selected, else those from step (ii) remain. **(iv)** Collinearity between variables is checked. The linear correlation between the indices selected at each step is used to calculate the Variance Inflation Factor (VIF) (Freund et al., 1998). If this exceeds a threshold (here chosen to be 5.0), indicating the predictors are highly correlated, the variable adding least skill to the model is removed. Using a more restrictive threshold of 2.0 for the VIF does not change the conclusions drawn from the results presented in this article. **(v)** Steps (ii) to (iv) are repeated until the addition of a predictor increases the explained variance by less than 1% or until all the predictors have been included already. The same approach has been used to model the monthly variability of daily average $PM_{2.5}$ and 1500 UTC ozone in Section 5.

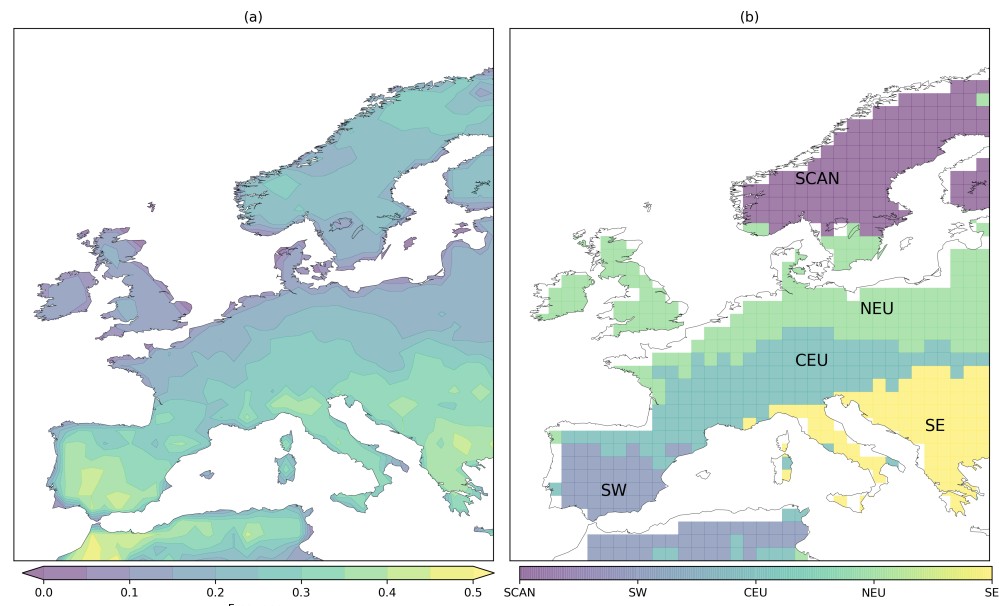

**Figure 1.** (a) The annual of frequency of air stagnation in Europe between 1979 and 2018 in the ERA5 data set. (b) Regions within Europe identified as having distinct air stagnation characteristics. These regions have been named Scandinavia (SCAN), northern Europe (NEU), central Europe (CEU), southwest Europe (SW) and southeast Europe (SE), following Garrido-Perez et al. (2018).

## 3  The large-scale circulation during air stagnation

The climatological frequency of air stagnation in the ERA5 dataset is shown in Figure 1. In Europe, air stagnation is most common in the Mediterranean region (plus northern Africa) with annual stagnation frequencies in the region around 40%. Frequencies then reduce polewards (values of 15% for the UK and across north-central Europe) before increasing again in Scandinavia where stagnation frequency is near 30%. Five regions within Europe with distinct air stagnation characteristics were identified by clustering the monthly frequencies of air stagnation as in Garrido-Perez et al. (2018): Scandinavia (SCAN), northern Europe (NEU), central Europe (CEU), southwest Europe (SW) and southeast Europe (SE). The regions are depicted in Figure 1 (b).

### 3.1  Annual air stagnation events

In this section, composites of the large-scale circulation are shown for air stagnation events occurring in selected regions. Air stagnation events are defined as occasions when stagnation occurs in at least half of the grid points within a region for at least four consecutive days. A threshold of four days was also used to define stagnation events in Wang and Angell (1999) and Huang et al. (2017). Composites are shown for two regions, SW and NEU, as examples of the distinct effect of the large-scale

circulation on stagnation in Southern and Northern Europe. Note that in the case of subtropical ridges, the algorithm described in Section 2.2 detects them in three longitudinal sectors (ATL, EUR and RUS). In the following, they will be considered over the three stagnation regions where they may be found (SW, CEU and SE), because by definition they do not extend northward to cover NEU and SCAN. Statistical significance is assessed for the composites by means of a Monte Carlo, bootstrapping approach. The composite value obtained under the condition of stagnation at each grid point is compared with 5000 randomly generated composites of the same size drawn from the climatology.

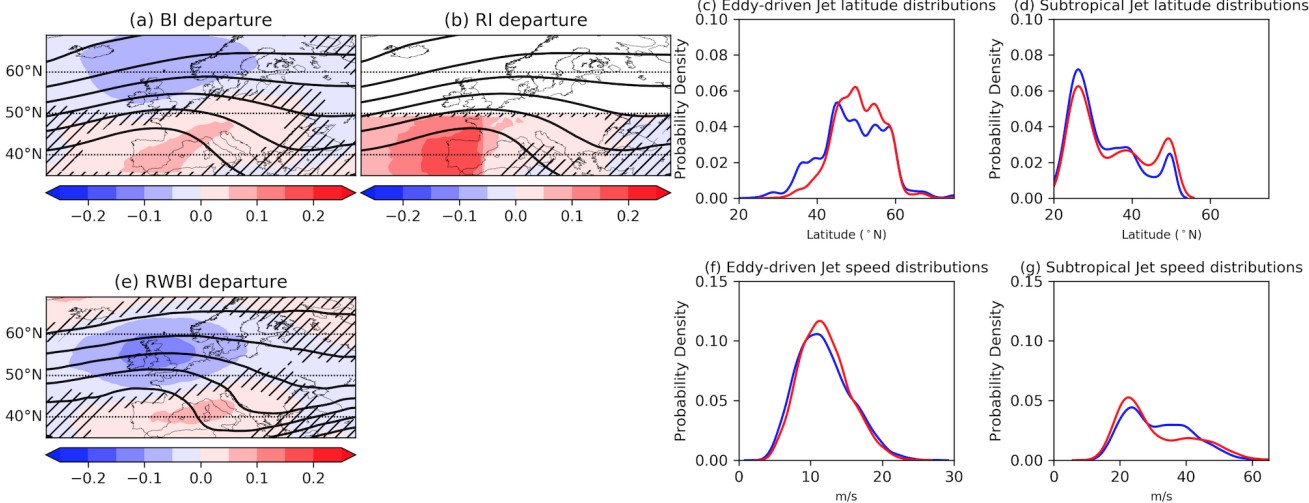

**Figure 2.** (a) Blocking index departure (shading) and composite Z500 (line contours), (b) ridge index departure (shading) and composite Z500 (line contours), and (e) Rossby wave breaking index departure (shading) and composite $\theta_{2PVU}$ (line contours) for stagnant days over SW. In (a), (b) and (e) the blocking, ridge and Rossby wave breaking index frequencies are presented as departures from their annual climatological frequencies, respectively. (c) The climatological annual eddy-driven jet stream latitude distribution (blue) and jet latitude distribution during stagnant days (red). (f) As in (c) but for the speed of the eddy-driven jet stream. (d) The climatological annual subtropical jet stream latitude distribution (blue) and that during stagnation (red). (g) As in (d) but or the subtropical jet speed. Hatching in panels (a), (b) and (e) denotes where there is not statistical significance (p > 0.01) calculated using a Monte Carlo approach (see text). Data source: ERA5 reanalysis during 1979-2018.

Annual mean composites of the large-scale circulation for air stagnation events in SW are shown in Figure 2. When stagnation occurs in SW, there is often a subtropical ridge extending into southwestern Europe. This is evidenced by an ∼20% increase in the climatological frequency of subtropical ridging over the region (Fig. 2 (b)). The ridging pattern present in the composite $Z500$ field (Fig. 2 (a), (b)) can sometimes exhibit a flow reversal in midlatitudes, resulting in a slight increase in block frequency over the region. Rossby wave breaking is less frequent to the north of SW during stagnation, consistent with the upper-level jet being centred over this region. Both jets are more frequently observed around 50°N during stagnation, which likely reflects that the two can merge north of the SW region. Specifically, the eddy-driven jet is less frequently in its southern

mode during SW stagnation (red distribution in Fig. 2(c)) and tends to be located further north. The subtropical jet latitude distribution during stagnation is more similar to its climatology, with only a slight increase in its northern flank and decrease in its southern latitudes. The speed of the eddy-driven jet is increased when stagnation occurs in SW, which is a robust feature associated with subtropical ridges (Sousa et al., 2018), while the subtropical jet speed is reduced. Note that the intensification
and weakening of the eddy-driven and subtropical jets, respectively, occur at different latitudes and longitudinal sectors. The stagnation speed and latitude distributions of both jets are significantly different from their climatologies (p<0.01, two-sample Kolmogorov-Smirnov (K-S) test). The composites of each dynamical index are similar for SE stagnation, but with the increased frequency of ridges shifted to the east (not shown).

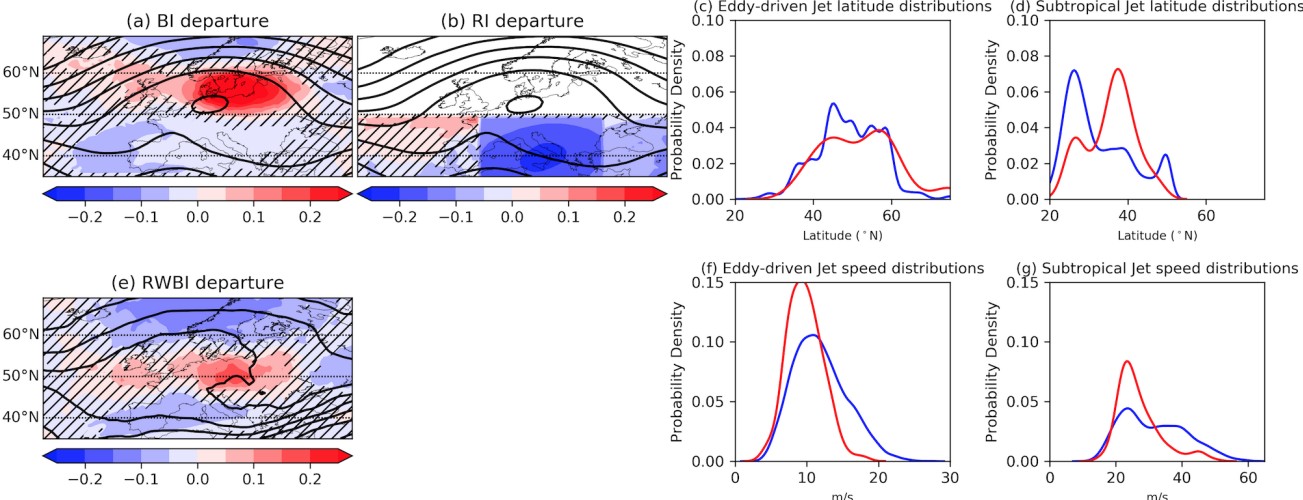

**Figure 3.** As in Figure 2 but for stagnation in NEU.

In Figure 3, composites of the large-scale circulation for stagnation in NEU are shown as a representation of northern Europe
(SCAN, NEU and CEU). Stagnation occurring in northern regions is associated with a large increase in block frequency over the stagnant area, seen also as an amplified large-scale wave in Z500 (Fig. 3(a), (b)). For NEU, it extends from the UK across most of Europe and a cut off region of high pressure is evident at its centre with the increase in block frequency exceeding 25%. The flow reversal associated with this blocking pattern also involves lower pressures and hence a strong reduction in subtropical ridge frequency south of the block structure (Fig. 3 (b)). The large-scale, upper-level wave is also seen in the
mean potential temperature field on the dynamical tropopause, which features an anticyclonically breaking Rossby wave. This corresponds to an increase of anticyclonic RWB slightly south of the region of increased block frequency. The eddy-driven jet is less frequently in its central mode (Fig. 3 (c)) as it would typically result in strong winds not conducive to stagnation over the regions, and shows the largest departures in northern latitudes (poleward of 60°N). Differently, the subtropical jet latitude exhibits a strong increase in the central latitudes of its distribution (around 40°N), reflecting the southern branch of the
split jet pattern associated with the block. Both jet streams are also significantly less intense during NEU stagnation (p<0.01,

two-sample K-S test): the distributions of jet speed are both notably shifted toward lower wind speeds (Fig. 3 (f), (g)). Blocking and RWB are also more frequent for stagnation in SCAN and CEU and are also associated with amplified upper-level waves centred over the regions (not shown). The reduction in ridge frequency during stagnation in NEU is present for SCAN but not for stagnation in the more equatorward latitudes of CEU. For stagnation in CEU and SCAN, the eddy-driven jet is less frequently over the latitudinal ranges encompassed by these regions (the southern and northern modes, respectively), and the subtropical jet index captures the southern branch of the blocking-induced split jet pattern in both regions.

Summing up, the dynamical signatures of stagnation as inferred from the composited departures of the dynamical indices are larger for northern regions in Europe. The comparatively weaker anomalies for the southern regions reflect the fact that the mean weather conditions in southern Europe are more favourable (drier and less windy) for stagnation. This does not however mean that stagnation in southern regions is less related to the large-scale circulation. Finally, the composites shown in Figures 2 and 3 are for stagnation defined by the Horton et al. (2012) index. Repeating the analysis using the Wang et al. (2018) and Huang et al. (2018) indices gives very similar results (not shown) and suggests that the large-scale circulation signatures highlighted are indeed important for stagnant air occurrence and not a feature of the chosen ASI. The composites of the large-scale flow are also similar when considering stagnation in each season separately (not shown), suggesting that stagnation occurs under similar set ups of the large-scale flow in all seasons. The seasonality of the large-scale connection to stagnation is further explored in the remainder of this section.

### 3.2 Seasonal and lag dependence of the large-scale circulation

Stagnation events occurring year-round have been examined so far. Now we explore the seasonality and time dependence of the relationship between the large-scale dynamics and air stagnation. To do this, we define a metric quantifying the correspondence between stagnation and selected dynamical indices. We term this the *departure measure* and define it as

$$\frac{1}{N_R} \sum_{i \in R} |I_{stagnation} - I_{climatology}|, \tag{2}$$

where $R$ is the region, $i$ a grid point within region $R$, $N_R$ is the number of grid points in the region and $I$ is the dynamical index. For simplicity, this subsection excludes large-scale dynamical factors (i.e. jet streams) and focuses on the most immediate regional drivers (block, RWB and ridge index-frequency), for which lagged relationships can be better interpreted. The departure measure is thus unitless. The Rossby wave breaking index is included for all regions while the block and ridge indices are shown only for regions where they are most influential (northern and southern regions, respectively; Section 3.1). In addition, the four day persistence criteria is removed here to ensure enough stagnant days are included in the analysis, which is necessary when considering northern regions in winter.

Departure measures as a function of lag about stagnant day are shown in Figure 4, for each region, season and dynamical index. There are three features to highlight in Figure 4. Firstly, the large-scale dynamics and stagnation correspondence tends to be strongest in winter (departure measure is largest), except for ridges in SE which peak in summer (Fig. 4 (e)) and blocks in SCAN which peak in spring (although with departure measures very close to those in autumn and winter, Fig. 4 (a)). Air stagnation is more closely related to the large-scale circulation in winter because calm, dry conditions are rarer at this time of

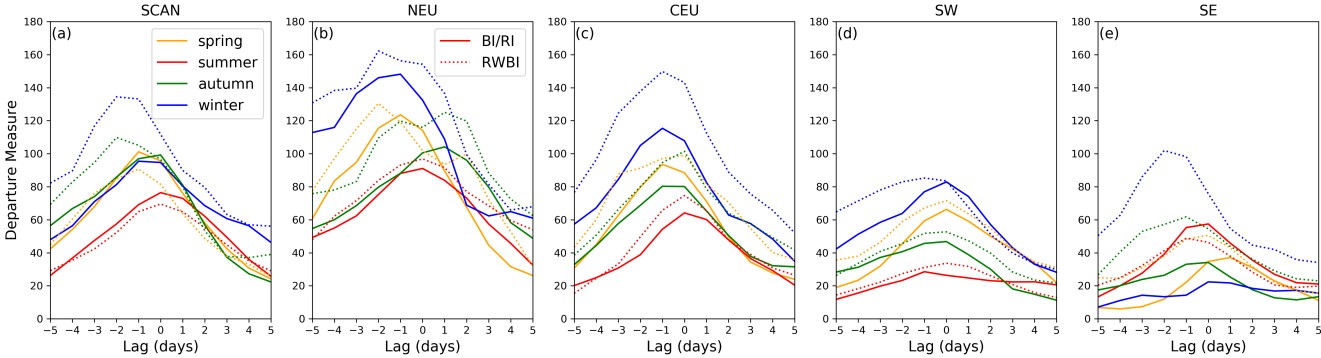

**Figure 4.** Departure measures for the blocking or ridge index (solid lines), and the Rossby wave breaking index (dotted lines) during stagnation for each season as a function of lag from stagnant days in (a) SCAN, (b) NEU, (c) CEU, (d) SW and (e) SE. The blocking index is shown for SCAN, NEU and CEU and the ridge index for SW and SE.

year, and need to be driven by a synoptic-scale anticyclone more frequently. Secondly, the departure measures for SCAN, NEU and CEU are higher than in SW and SE, involving larger departures of the atmospheric circulation. This again relates to the less favourable weather conditions for stagnation in northern regions, which means that larger anomalies are required therein for stagnation to occur, as compared to southern regions. Finally, the large-scale dynamics leads the changes in air stagnation by a few days. This can be inferred from the peak in departure measure for negative lags. This lag between the large-scale dynamics and air stagnation could have implications for the predictability of air stagnation.

### 3.3 Stagnation and pollution response to large-scale drivers

In addition to analysing how the dynamical indices depart from the climatology on stagnant days, the reverse situation can also be considered, i.e. how stagnation differs on days that are blocked, for example. To do this, the likelihood of a stagnant day occurring is compared in the climatology (including all days) to days when a specific driver dominates the region. We can also test the response of the pollutants to these drivers. Defining an extreme pollution event as a day on which the mean pollutant level in a region is above its $90^{\text{th}}$ percentile, the frequency of extreme events can be compared on days that have a large-scale driver dominating the region. Extreme days are defined for each season separately for both pollutants to account for their seasonal cycles. The change in likelihood in the occurrence of stagnant days and extreme pollution days are shown for each region and season in Figure 5. Changes are shown for when blocking occurs in all regions as well as for subtropical ridges in SW and SE. Rossby wave breaking occurring in a region is found to have a smaller impact on the likelihood of stagnation or extreme pollution, particularly for southern regions (not shown), so is omitted here. So whilst Rossby wave breaking is more frequent than normal during stagnation in these regions (Fig. 4 (d), (e)), the occurrence of Rossby wave breaking in the region does not change the likelihood of stagnation.

Concerning the specific regional drivers, overall, air stagnation is around twice as more likely to occur when there is a block or ridge present, though this number is sensitive to the precise definition of a region being blocked or ridged (not shown).

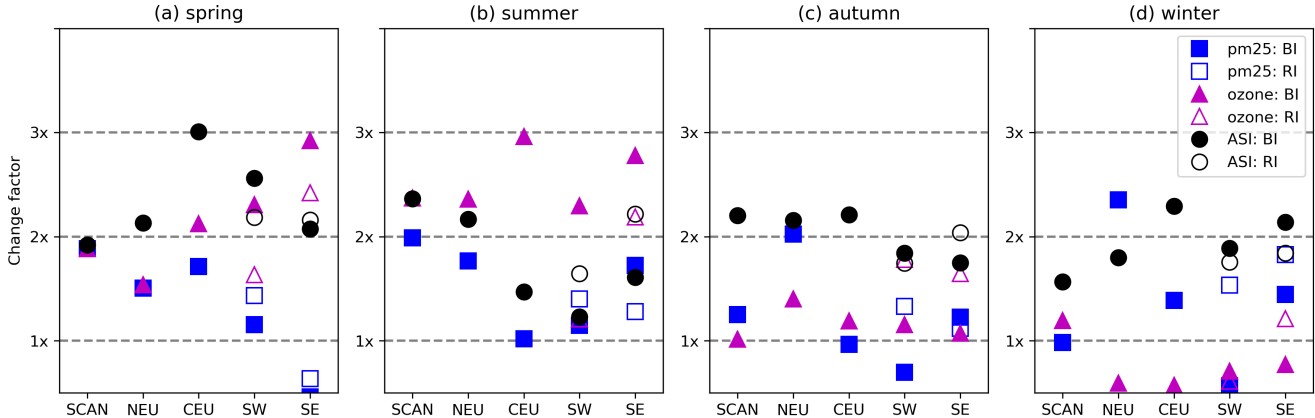

**Figure 5.** Change in the likelihood of stagnant days (ASI), extreme ozone days, and PM$_{2.5}$ days under the presence of blocking (BI, shown for all regions) and subtropical ridges (RI, shown only for SW and SE) in (a) spring, (b) summer, (c) autumn and (d) winter. Dashed lines at 1x, 2x, and 3x represent no change, a doubling and a tripling in likelihood, respectively.

This doubling in the likelihood of stagnation occurrence for a given driver is consistent for stagnation in SCAN and NEU during every season, in SW and SE during spring, autumn and winter, and CEU in autumn and winter (Fig. 5). The response of stagnation to the large-scale drivers is smaller for southern regions in summer (Fig. 5 (b)) and larger for CEU in spring (Fig. 5 (a), when the likelihood is tripled). In southern regions during summer, when the influence of the drivers is small, stagnation is most frequent (over 50% in the climatology) as conditions favourable for stagnation are common and can occur without a large-scale driver. A ridge being present in these regions does still increase the occurrence of stagnation (on average, around 75% of ridge days are also diagnosed as stagnant).

The dynamical drivers also increase the likelihood of extreme pollution events, though the results are more complicated due to the properties of the pollutants and how they respond differently depending on the season and region. Extreme ozone days are more than twice as likely to occur in the majority of regions under the influence of blocking in spring, and particularly summer (Fig. 5 (a), (b)). Extreme ozone events are around three times more likely to occur when there is a block in SE in spring and summer, as well as in CEU in summer. Calm conditions and high temperatures, such as those found during blocking in these seasons, result in increased ozone levels. Extreme ozone events also occur more frequently when a subtropical ridge is identified over SW and SE, particularly in SE when their likelihood is approximately than doubled. Extreme ozone days are less likely to occur during a blocking event in winter (Fig. 5 (d)) in all regions apart from SCAN (where the change is near zero). In winter, the stable situation during a block leads to ozone reduction, because it is lost by reaction with nitrous oxide and dry deposition in a shallow boundary layer. There is little change in ozone extremes during blocks in autumn (Fig. 5 (c)), though increases are evident during subtropical ridges in SW and SE. Conditions during subtropical ridges in southern Europe in autumn remain sunny and warm and favourable for ozone production. The different response of this pollutant to blocking in winter compared to other seasons is consistent with the findings of Ordoñez et al. (2017). The large-scale influence

on PM$_{2.5}$ is more regionally dependent. PM$_{2.5}$ extremes are more frequent during blocking in NEU, particularly in autumn and winter (Fig. 5 (c), (d)), as well as in SCAN during spring and summer (Fig. 5 (a), (b)), with extreme days around twice as frequent. Extreme PM$_{2.5}$ days are also more likely when a block or ridge dominates in CEU in spring and winter, and SW during summer and winter, with little change in those regions in the other seasons. There is also generally little change in the occurrence of extreme PM$_{2.5}$ days when a subtropical ridge or block dominates in SW. This may suggest stagnation is a better

predictor of ozone than PM$_{2.5}$ in these regions. The components that constitute PM$_{2.5}$ may differ in each region and will behave differently under the influence of the synoptic-scale weather systems and, as such, respond differently to their occurrence. We now explore to what extent the large-scale circulation can explain the variability in air stagnation, followed by a comparison with that explained for the pollutants.

## 4  Modelling the variability of European air stagnation

In this section, the MLR method described in section 2.3 is used to model the monthly variability of air stagnation. The monthly count of stagnant days in each region and season is modelled separately using the dynamical indices as predictors. A day is counted as stagnant if the number of stagnant grid points in the region is above its 50th percentile in the climatology. The results presented in this section are robust to the choice of this threshold. The dynamical indices are used as predictors as follows. The blocking (BI), Rossby wave breaking (RWBI), and ridge (RI) indices are regionalized as the number of days in a

325 month when the region was dominated by such weather systems, which occurs if the number of affected grid points is above its 50th percentile. As mentioned above, ridges are only considered for CEU, SW and SE. The monthly mean latitude and speed of the eddy-driven (JL and JS) and subtropical (STJL and STJS) jets are used as direct predictors (the time series of these predictors are the same for every region). Recall that we only use regional or large-scale predictors (based on $Z500$, wind speed throughout the troposphere, and $\theta$ on 2 PVU) to quantify the percent of variance in stagnation that can be explained from

330 the dynamics, without including smaller-scale or local phenomena, such as local winds or convection. We exploit the stepwise procedure to identify the large-scale dynamical processes most related to air stagnation as follows.

The stepwise MLR models are constructed for each region and season. All regions share the jet-based predictors, as they are defined independently of the region (i.e. Europe-wide). For the remaining drivers (BI, RWBI, RI), each region has its own set and time series of predictors that account for regional information on the frequency of occurrence of synoptic-scale systems,

with RI being included only for those regions where ridges occur (SW, SE and CEU). For example, we model the monthly variability of air stagnation in SCAN (AS$_{SCAN}$) as

$$\mathrm{AS}_{SCAN} \approx \beta_0 + \beta_1 \mathrm{BI}_{SCAN} + \beta_2 \mathrm{RWBI}_{SCAN} + \beta_3 \mathrm{JL} + \beta_4 \mathrm{JS} + \beta_6 \mathrm{STJL} + \beta_6 \mathrm{STJS}, \qquad (3)$$

where BI$_{SCAN}$ and RWBI$_{SCAN}$ are the monthly time series of blocked days in SCAN and RWB days in SCAN respectively. $\beta_0$ is the model intercept and the $\beta_i$ terms are the regression coefficients.

In addition, for each region we construct a second model that also incorporates the regional predictors defined for the remaining regions. By including the time series of BI, RWBI and RI for all regions we can account for remote influences of

the atmospheric circulation on air stagnation. These synoptic systems can have a remote (non-local) impact on stagnation by influencing the weather conditions both upstream and downstream of their location. Therefore, the second model expands the set of predictors by considering those that are specific of the other regions. Using stagnation in SCAN as an example again, this second model is constructed as

$$AS_{SCAN} \approx \sum_{regions} BI_R + \sum_{regions} RWBI_R + \sum_{regions} RI_R + JL + JS + STJL + STJS, \tag{4}$$

where subscript $R$ refers to the region of the index time series included. As in equation 3, the model includes an intercept and regression coefficients but they are omitted here for clarity. For both models, the stepwise procedure is subsequently applied to select the leading predictors and avoid the inclusion of predictors that are too highly correlated with one another. We term the first model (equation 3) the regional model and the second (equation 4) the European model for the remainder of this section.

## 4.1 Variance in stagnation explained by the dynamical predictors

The coefficients of determination ($R^2$) for the regional and European stepwise MLR models are presented in Figure 6. Between 30 and 50% of the monthly variability of air stagnation can be explained by the regional models in most cases, though this is dependent on region and season. For instance, more than 50% of the variance in stagnation in NEU can be explained by the regional model for most seasons, whereas it is less than 25% for SCAN in spring. This increases to between 40 and 70% when including remote predictors in the regression, highlighting the faraway effect weather systems, such as blocks and ridges, can have on the surface weather. Both model set ups give a statistically significant relationship between air stagnation and the large-scale circulation for all regions and seasons (p<0.01). The model skill is generally highest in winter, consistent with the closer association between stagnation and the large-scale circulation in winter shown in section 3. The model does not perform better in northern or southern regions in Europe: model skill is similar in SW, CEU and NEU and generally lowest in SCAN and SE. This suggests that weaker circulation anomalies can have a comparatively larger effect in southern compared to northern regions, such that smaller departure measures (Fig. 4) in southern regions can lead to $R^2$ values as high as those in northern regions.

Comparing the explained variance between the two models gives insight into which seasons and regions are most affected by the remote drivers. Overall, the remote predictors compensate for the lower skill of local predictors in SCAN and SE in most seasons ($R^2$ increases greater than 0.15). They add comparatively lower skill in NEU and CEU in the cold seasons and SW in the warm seasons. The variance in stagnation explained by the European models in these regions is more consistent across seasons than that obtained from the regional models. As regional predictors are favourable for stagnation in the region they occur, we hypothesize that the remote predictors most likely bring model improvements by informing the occasions when stagnation is disfavoured in the region, i.e. they have a negative coefficient in the MLR equation.

## 4.2 Dynamical predictors of air stagnation

The predictors selected by the stepwise regression method are discussed in this section for each region and season. We now only consider the European model that uses both local and remote predictors as it consistently outperforms the regional model. The

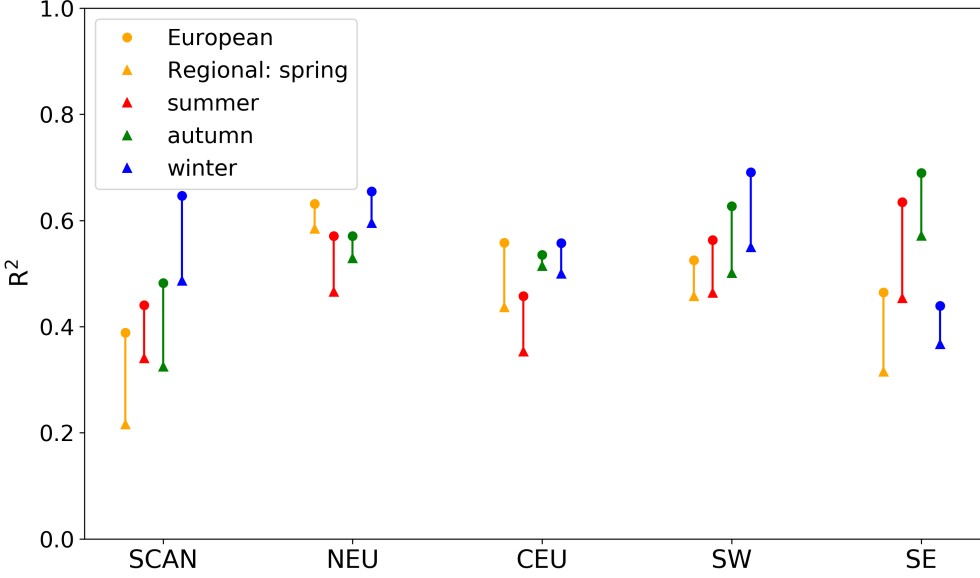

**Figure 6.** Skill of the stepwise multiple linear regression models ($R^2$) of monthly air stagnation in each region and season (colours) during 1979–2018. Skills are shown for the regional models (triangles) and the European models (circle). See text for an explanation of the regional and European models.

predictors selected by the model represent the dynamical drivers that are needed to explain the most variance in air stagnation.

There are a total of 17 predictors for the stepwise method to select from: five regional BI and RWBI predictors (one for each region), three RI predictors (for CEU, SW and SE), together with the jet latitude and speed time series for the eddy-driven and subtropical jets. The model skill ($R^2$) achieved by the selected predictors are shown for each season and region in Figure 7. The model skills shown in the maps correspond to the $R^2$ values reached after adding the final predictor into the model.

First we detail some statistics about the selected large-scale predictors for air stagnation. The number of predictors necessary

to include in the MLR model of air stagnation varies between 4 and 12, depending on the region and season. The average number of predictors included in the model is about 7 for SCAN and NEU and 9 for CEU, SW and SE. Therefore, fewer predictors are typically required to explain stagnation variability in northern regions. The colours behind each included predictor show the model $R^2$ as each of the predictors is added (from left to right) in the formulation. In some cases, one predictor can explain a large fraction of the variance, for example in NEU winter stagnation, and the additional predictors add relatively little skill

to the model (though still more than the 1% threshold required by the stepwise formulation). Other times, more predictors have a more equal contribution to the overall model skill, for example when describing stagnation in SW winter. This does not necessarily impact the overall skill of the model, which is in both cases above 60% of monthly stagnation variability.

The leading predictors for regional air stagnation are most often RWB and blocking (each 7 times out of the total 20 region/season combinations), with ridges and the subtropical jet speed being the leading predictor in 3 combinations each.

Many regions and seasons include both RWBI and BI as predictors (in a few cases from the same regions). Despite their

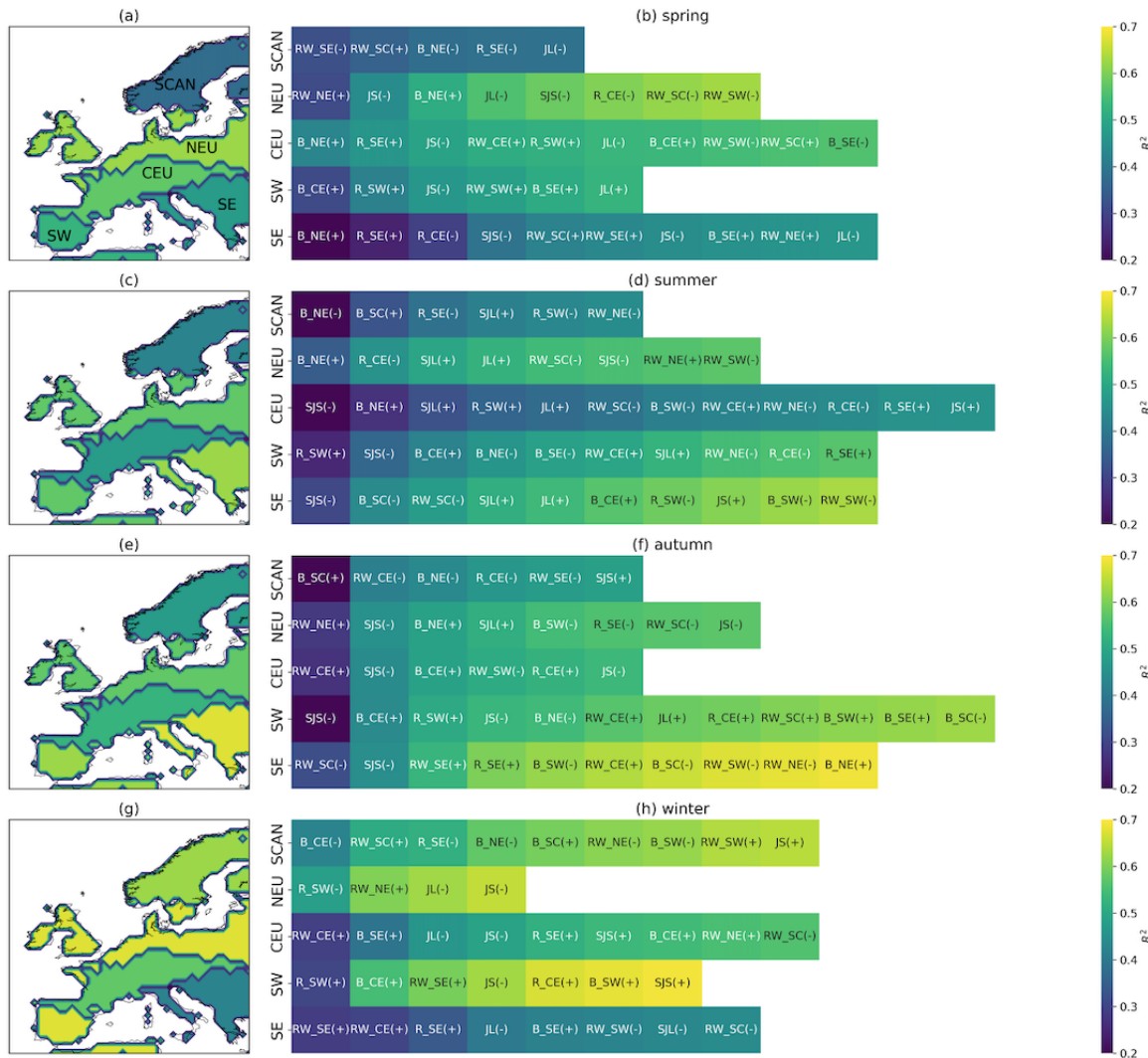

**Figure 7.** Model $R^2$ for stagnation occurring in (a) spring, (c) summer, (e) autumn and (g) winter. Predictors included in the stepwise linear regression for each region in (b) spring, (d) summer, (f) autumn and (h) winter, where the plus and minus signs for each predictor represent whether the coefficient in the MLR model is positive or negative, respectively. Colours in the right hand panels show the $R^2$ of the model as each predictor is added in its formulation. The $R^2$ values are calculated for monthly data in 1979–2018.

apparent similarity, these two indices can reflect different block configurations and persistent signatures, as stated above, which may partly explain this result. Out of the 20 leading predictors 12 have a positive contribution to the model (8 have negative) and 9 are local processes (11 remote drivers), although these figures are all dependent on the region and the season considered. We recall that in many cases the difference in explained variance between a given predictor and the following one

is small. Therefore, the specific order in which predictors are selected may not be necessarily meaningful from a dynamical point of view.

    In spite of these limitations, we can use the stepwise regression results to make some inferences on the dynamical influences. In SCAN, RWB and blocking occurring locally dominate stagnation variability. Only in winter are both predictors included by the model, though, suggesting the indices are identifying different features in this season. BI and RWBI occurring in

other regions (south of SCAN) inform stagnation reduction (negative coefficient), because they are associated with a poleward strengthening of zonal winds and hence increased wind speeds over SCAN. Subtropical ridges in SE and CEU have similar effects (e.g. Sousa et al., 2018), in agreement with their negative contribution to stagnation frequency. The eddy-driven jet is not identified as an important driver for stagnation in SCAN (only appearing twice as the final predictor selected in the model) supporting the results of Madonna et al. (2017) who found that variations in Scandinavia blocking and the eddy-driven jet are

relatively unconnected. The poleward strengthening of the zonal wind associated with blocks and ridges occurs outside the region of the eddy-driven jet index. The subtropical jet is also not important for stagnation in SCAN, which is expected as this region is too far north.

    Local blocking or RWB also drive stagnation in NEU, appearing as the leading predictor in every season except winter (when it appears second). Winter stagnation in NEU is neatly described using only four predictors. It is favoured by local RWB

and reduced when the eddy-driven jet is strong and located over this region, as inferred from the negative terms for JS and JL, as well RI over SW. On the other hand, subtropical ridges reduce stagnation in NEU in the other seasons by shifting the jet stream over the region. Indeed, the eddy-driven jet is an important predictor for NEU across all seasons, as this region lies at the jet exit region and will be more directly impacted by its characteristics.

    In CEU, both local and remote occurrences of blocking and RWB influence stagnation occurrence. This region covers a

relatively small region meridionally, smaller than the typical meridional scale for a blocking event, so a block situated over NEU or SE would also be expected to extend over CEU and favour stagnation therein. Blocking and RWB in CEU, NEU and SE are thus typically positive predictors in the model, with the exception of Rossby wave breaking over NEU in summer when it only appears as the ninth predictor. RWB in SCAN can have a positive or negative influence on stagnation in CEU, depending on the season (although the amount of variance explained by this predictor is often small). This behaviour could

reflect seasonal changes in the latitude of RWB or in the associated pattern. For example, a dipole-type block over SCAN would be associated with a cyclonic circulation anomaly over CEU and hence increased wind, and a reduction in stagnation likelihood. An omega-type block may extend across parts of both SCAN and CEU and bring settled anticyclonic conditions to both regions. Stagnation in CEU is typically reduced when the eddy-driven jet speed is increased. The opposite is true in summer, when the eddy-driven jet shifts north towards latitudes farther from CEU. Local ridges are not as important for

stagnation in CEU, appearing further down the list of predictors, arguably because they involve large meridional excursions form their subtropical sources and tend to break, therefore being better described by the RWBI or BI.

    In contrast to CEU, subtropical ridges are key for stagnation variability in SW, appearing twice as the leading predictor and once as the second included predictor. They bring sunny, settled conditions over SW and can thus support stagnation occurrence (e.g. Santos et al., 2009). The so-called low-latitude blocking events (i.e. those occurring in SW and SE) can be

easily confounded with subtropical ridges (Sousa et al., 2018), which sometimes expand over vast subtropical regions and hence can also promote stagnation in SW. Blocking systems centered polewards of SW, such as those in CEU, can also extend over their anticyclonic influence over SW depending on their location, scale and shape. The subtropical jet speed is a key predictor for SW stagnation in summer and autumn when a weaker subtropical jet favours stagnation occurrence. In winter, the subtropical jet is situated further south than the SW region so it will not directly impact stagnation therein (it appears as the last predictor, with positive sign and adds a small amount of explained variance). Increases in the eddy-driven jet speed also predict a reduction in SW stagnation for all seasons except summer, when the eddy driven jet is located at its northernmost latitudes.

Although local blocking and RWB also tend to dominate stagnation variability in SE, the dynamical predictors are more numerous and seasonally-varying in this region than in the others. A strong subtropical jet reduces stagnation in all seasons but winter. Highlighting a similar weather pattern, local ridges are associated with increased stagnation in most seasons. In spring, when blocking reaches large areal extents, the leading predictor of SE stagnation is blocking in the neighbouring region of NEU. As the SE region is farther downstream of the North Atlantic, the eddy-driven jet speed and latitude are not identified as important drivers of regional stagnation.

Thus far, we have given a broad explanation of some of the key predictors of stagnation in each region. A complete explanation of the predictors related to stagnation in every region and season is unfeasible, so, as an example, air stagnation occurring in SW during winter is explored further, which is the combination yielding the highest skill. The leading five predictors are included in our discussion. They are local subtropical ridges, blocking in CEU, RWB over SE, the eddy driven jet speed, and subtropical ridges over CEU. To aid in the explanation two measures are considered. Firstly, the partial correlation in the MLR model between the stagnation time series and each predictor is calculated. The partial correlations are similar for each of these predictors and so none of the large-scale features dominate over the others. Secondly, composites of air stagnation frequency anomaly are produced for the four selected predictors that are defined regionally. The composites are produced by comparing stagnation occurring on days when each predictor index covers half the region to the climatological frequency of stagnation (the same method for the composites shown in Section 3), and are shown in Figure 8. Whilst these composites do not reflect the linear behaviour of the MLR model, they do demonstrate the direct impact the predictors have individually on the occurrence of air stagnation and are useful when interpreting the stepwise model results.

Subtropical ridges identified in SW and the contiguous CEU region are both associated with increased stagnation frequencies in SW. The associated anomalously high geopotential height in these regions promotes anticyclonic circulation and settled conditions with suppressed rainfall, which favour stagnation. The influence of SW ridges is locally restricted whereas ridges in CEU promote stagnation across the Mediterranean (Fig. 8 (a), (d)). SW ridges are likely related to those defined as Atlantic ridges in Sousa et al. (2018) and hence are associated with stronger winds north of SW, where stagnation is reduced. Ridges identified in CEU will also include European ridges in the definition of Sousa et al. (2018) associated with high geopotential heights across the region. Blocking in CEU has a similar effect (Fig. 8 (b)). Blocks in this region often develop from subtropical ridges extending from lower latitudes and are identified by the block index when the ridge begins to overturn. This typically begins anticyclonically on the eastern flank of the ridges and hence block onsets in this region are typically identified as anticyclonic (Masato et al., 2012). This means that the area to the south and east of the block remains in a region of anticyclonic

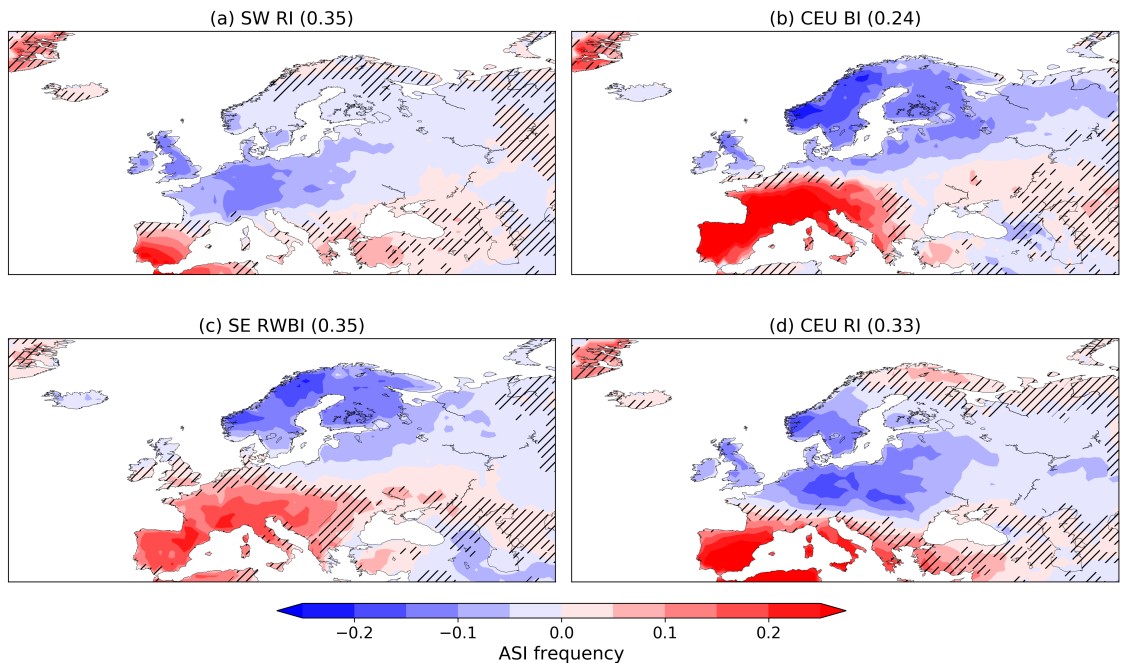

**Figure 8.** Composite departures (colour shading) and partial correlations (numbers in brackets on top of panels, based on monthly data) of stagnation occurrence in winter for days with (a) subtropical ridges in SW, (b) blocking in CEU, (c) Rossby wave breaking in SE and (d) subtropical ridges in CEU. These are among the five leading terms in the stepwise model for stagnation in SW winter. Hatching again denotes where there is not statistical significance (p>0.01) calculated using the approach described in Section 3.1.

circulation with conditions favourable for stagnation as the ridge develops. This feature is evident in the composite of block frequency for SW stagnation (Fig. 2 (a)) as an increase in block frequency extending from SW into CEU with a SW-NE orientation. The fourth predictor has a negative regression coefficient and describes the eddy-driven jet speed. Strong winds are obviously not conducive for stagnation occurrence, particularly when the jet is located in its southern mode which is directly above SW (Fig. 2 (c)). SW stagnation is also more frequent on days defined as having Rossby wave breaking in SE (Fig. 8 (c)). Anticyclonic RWB can occur over SE as a result of the regional meridional shear of the zonal wind (i.e. when mid tropospheric winds intensify to the north of the region and/or weaken over SE). This is expected to enhance the SE-NE tilt of the eddy-driven jet, diverting cyclone tracks away from southern Europe and the Mediterranean. On the other hand, RWB in SE can also be a regional signature of anticyclonic RWB on the eastern flank of a large-scale European block extending its anticyclonic influence (weak winds and absence of precipitation) towards SW. The large-scale circulation pattern would resemble that in Figure 3 (e), but shifted to the south.

In this section, synoptic-scale weather systems and features of the large-scale circulation have been shown to be important for describing the variability in air stagnation in the Horton et al. (2012) ASI, and specific regional drivers have been highlighted

as key predictors in specific regions and seasons. This choice of ASI has been motivated by the known influence of the index on air pollution in Europe (e.g. Garrido-Perez et al., 2018, 2021), and the availability of long-term meteorological time series that can be used to compute the ASI as compared to the relatively short air quality datasets. As discussed in the introduction, however, previous studies have shown that results may be sensitive to the choice of ASI and that stagnation may not always be a suitable proxy for pollution events (e.g. Oswald et al., 2015; Kerr and Waugh, 2018). We test the robustness of the results obtained using the Horton et al. (2012) ASI in the next section.

## 5  Comparison with other ASIs and pollutants

The European stepwise MLR is repeated in this section for modelling the monthly variability of air stagnation in the Huang et al. (2018) and Wang et al. (2018) ASIs, as well as the monthly variability in $PM_{2.5}$ and ozone. By doing this, we can assess how similar the synoptic- to large-scale influence is for the different ASIs and for direct pollutant datasets and verify that the results discussed in the previous section are robust and are relevant for air quality. We use the common time period covered by the pollutants and ASIs for all the models to allow a fair comparison (2003–2018). The monthly count of stagnant days in each region is used as the response variable for the ASIs, whereas the monthly mean pollution level averaged for each region is used as the response variable for $PM_{2.5}$ and ozone. Also note that the Horton et al. (2012) ASI includes the wind speed at 500 hPa in its formulation, and hence may be expected to be closely related to the large-scale circulation and synoptic-scale weather systems such as blocks, whereas the Huang et al. (2018) and Wang et al. (2018) ASIs contain no criteria based on the flow at upper levels and thus provide an additional test of the robustness of the large-scale correspondence.

The skill of the models for each ASI and pollutant variability are shown in Figure 9, for each region and season. Generally, the skill of the model is consistent across the ASIs and ozone ($R^2$ values normally between 0.5 and 0.8), though the model does worse for PM in most seasons and regions ($R^2$ values generally between 0.3 and 0.6). The overall consistency in model skill between the different ASIs implies that an ASI does not need a criterion based on the mid-tropospheric flow for the large-scale dynamics to be able to explain a significant amount of the variability, although the Horton et al. (2012) index outperforms the other indices in around half of the 20 possible combinations of regions and seasons. Furthermore, the additional consistency in the model skill between the ASIs and the pollutants shows that the large-scale circulation can explain much of the variability in pollutant levels in Europe as well, particularly for ozone. This adds to the evidence that there is a correspondence between air stagnation and ozone levels in Europe, although with some regional differences (Garrido-Perez et al., 2018, 2019). The model skill is somewhat lower for $PM_{2.5}$ in most regions and seasons, with minimum $R^2$ values of 0.30–0.50 in spring or summer and maximum $R^2$ values of 0.50–0.73 in winter (Fig. 9). Nevertheless, the MLR model shows a significant relation ($p<0.1$) between the dynamical predictors and $PM_{2.5}$ variability in all cases apart from spring in SW and SE and summer in NEU. The model skill for $PM_{2.5}$ is similar to that for the ASIs in winter, when stagnation has a large impact on atmospheric concentrations of this pollutant (Pandolfi et al., 2014; Garrido-Perez et al., 2021). PM is a mixture of many components, each having different sources and undergoing different processes, which means PM concentrations can be complicated to predict and may contribute to the lower model skill for this pollutant. It is beyond the scope of this paper to investigate aerosol-related

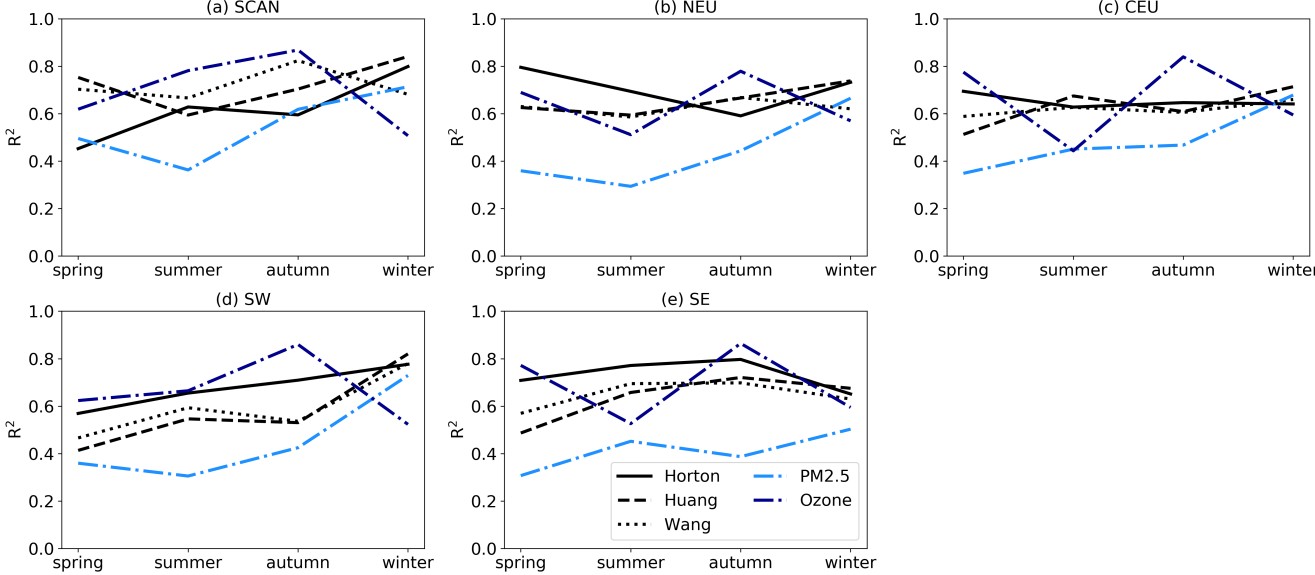

**Figure 9.** Stepwise multiple linear model skill in (a) SCAN, (b) NEU, (c) CEU, (d) SW and (e) SE for stagnation as defined using the Horton (solid), Huang (dashed) and Wang (dotted) indices and pollutants $PM_{2.5}$ (light blue) and ozone (dark blue) during the period 2003–2018.

processes and the impact of synoptic- to large-scale circulation on the different $PM_{2.5}$ components in the CAMS reanalysis. We have, however, demonstrated that large-scale influence is similar for air stagnation, ozone and wintertime $PM_{2.5}$. This adds to the evidence that ASIs are useful proxies for pollution levels in Europe and they can be used to aid our understanding of air quality and its variability.

To further explore how the large-scale circulation is related to the different stagnation indices and pollutants, the dynamical predictors of their variability selected by the stepwise procedure are compared. The most straightforward comparison is to examine all of the predictors selected by the stepwise procedure in the model of each ASI and pollutant for all seasons and regions combined. The fraction of times each predictor appears in the MLR model formulation is shown in Figure 10, for each ASI and pollutant. The dynamical indices related to air stagnation as defined in the three ASIs and those related to pollutant

variability are similar. In each case, the blocking and Rossby wave breaking indices are included most often as predictors with the indices for jet speed and latitude for both the eddy-driven and subtropical jet less so. This pattern is generally consistent when separating the selected predictors into regions and seasons, across the three ASIs and pollutants. However, there are some differences in the selected predictors for ozone and $PM_{2.5}$ in specific regions and seasons. For example, in SW during summer, a time when the model skill is considerably lower for $PM_{2.5}$ than for ozone (Fig. 9 (d)), subtropical ridges are more often the

predictors included for $PM_{2.5}$, rather than blocking that is selected for ozone. Blocking and subtropical ridges are important for $PM_{2.5}$ variability in SE during autumn, another occasion when the skill difference is large (Fig. 9 (e)), whilst they do not appear at all in the model of ozone, where Rossby wave breaking and the subtropical jet speed and latitude are most important. These differences highlight the complexity in modelling pollutant concentrations and how the response of the pollutants to

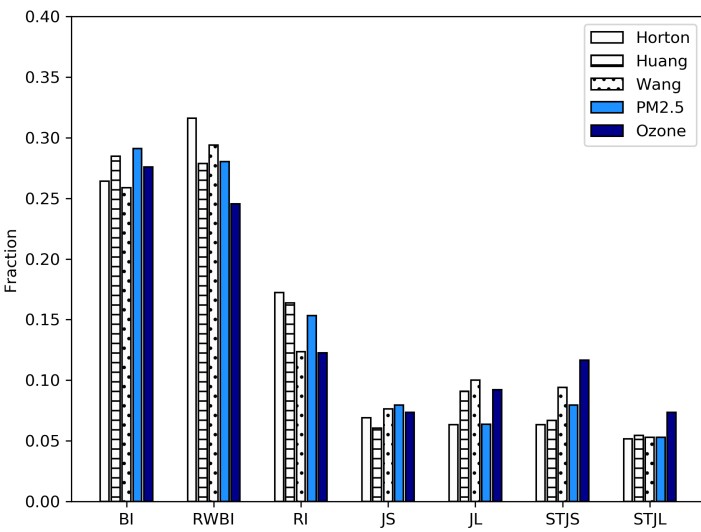

**Figure 10.** The dynamical indices included as predictors by the stepwise procedure in the MLR models of the three ASIs and two pollutants. The fraction of total predictors across all seasons and regions that are blocking (BI), Rossby wave breaking (RWBI), ridge (RI), eddy-driven jet speed (JS) and latitude (JL) and subtropical jet speed (STJS) and latitude (STJL) are shown for each model.

meteorological conditions can depend on the region and season. Nevertheless, the synoptic- to large-scale circulation remains
closely related to their variability in the majority of cases, and our findings have implications for studying stagnation and pollution variability in data from climate models.

## 6    Conclusions

Air stagnation can influence air quality (e.g. Jacob and Winner, 2009; Toro et al., 2019; Kanawade et al., 2020), especially in Europe (Garrido-Perez et al., 2018, 2021). As poor air quality is hazardous to human health (Pope III et al., 2002; Cohen
et al., 2017), it is beneficial to understand the processes causing air stagnation. Different local meteorological conditions have been shown to drive both air stagnation events and the build-up of specific pollutants in particular regions and seasons (Prtenjak et al., 2009; Dawson et al., 2014; Zhang et al., 2014). However, the large-scale circulation features driving such local conditions have comparatively been poorly addressed and are here used to describe air stagnation and air pollution characteristics within Europe. We find that synoptic-scale weather systems and features of the large-scale circulation are important for both air
stagnation and pollutant levels. The focus on air stagnation in the article, rather than pollution measurements directly, is motivated by the availability of long-term meteorological data and the simplicity of ASIs. They can easily be calculated from reanalysis or model data and hence allow for comparisons across many different time periods and regions across the globe. And whilst some studies have found inconsistent links between air stagnation and pollutants (e.g. Kerr and Waugh, 2018;

Garrido-Perez et al., 2019), their connection in Europe is generally strong (Garrido-Perez et al., 2018, 2021) and stagnation
can be used to provide information about pollutant levels.

In the present paper, air stagnation in Europe is shown to be strongly influenced by the large-scale circulation. Moreover, the large-scale leads changes in stagnation by a few days, which could have implications for its predictability. The large-scale circulation more often resembles a synoptic-scale high pressure system when stagnation is present. For northern regions this is achieved through the presence of an atmospheric blocking event whereas in southern regions it is more typically associated with
subtropical ridges. Indeed, subtropical ridges are unfavourable for stagnation in northern Europe, as they are associated with a poleward strengthening of winds which is unfavourable for stagnation and pollutant accumulation. Similarly, stagnation in Southern Europe is less frequent during high latitude blocking events (when strong winds and low pressure systems are diverted to the south of the block). These results are generally consistent with previous studies linking air stagnation or pollutant build up with large-scale meteorological drivers such as blocking and subtropical ridges (Garrido-Perez et al., 2017; Ordoñez et al.,
2017), the position of the midlatitude jets (Barnes and Fiore, 2013; Shen et al., 2015; Ordóñez et al., 2019; Kerr et al., 2020a,b), Rossby wave breaking (Webber et al., 2017), or cyclone frequency (Leibensperger et al., 2008; Tai et al., 2010, 2012; Leung et al., 2018), but our results provide a complete picture for stagnation events occurring year-round across Europe and allow discerning the relative roles of these multiple drivers.

Multiple linear regression models using a stepwise selection procedure have been used to model the monthly variability
of regional air stagnation in each region and season. Dynamical indices describing the large-scale circulation can be used to model the monthly variability of air stagnation and can explain around 60% of its variance. The model generally performs best when predicting stagnation in winter, particularly in Scandinavia. Winter weather conditions are typically windier and wetter than in other seasons so for stagnation to occur it is more likely to be driven by a large-scale circulation feature. This means stagnation and the large-scale flow are more closely related and the model skill is higher. Furthermore, surface weather
conditions are more strongly controlled by circulation changes in winter than in summer (Vautard and Yiou, 2009). The amount of variance explained by the large-scale circulation increases when including both local and remote predictors in the statistical model, particularly in cases when local phenomena are able to explain relatively little of the stagnation variability. Synoptic scale weather systems influence weather in upstream and downstream regions and are able to affect stagnation both locally and remotely.
The conclusions drawn from the results presented in this paper are robust to the choice of air stagnation index. The large-scale circulation shows similar signatures for the air stagnation events defined by three different indices, and is able to explain a similar amount of their variability, though for some regions and seasons the large-scale circulation shows stronger influences on the Horton et al. (2014) index, which also accounts for upper-level winds. Each of the stagnation indices considered here have been shown to identify situations in which pollutants can build up in various regions of the globe (Schnell and Prather,
2017; Garrido-Perez et al., 2018; Liao et al., 2018; Huang et al., 2018; Garrido-Perez et al., 2021). Indeed, the dynamical indices were able to explain a similar amount of monthly variability in ozone and wintertime PM to that in air stagnation. The predictors selected for the model were also similar in each case. Thus we have identified the synoptic- to large-scale drivers that are important for both air stagnation *and* pollution levels within Europe. This suggests that, despite their limitations, ASIs

can be used to understand air quality. In particular, the variance in air stagnation explained by the large-scale circulation can be used to understand pollutant variability, as well as changes in future high pollution episodes. This is important because ASIs can easily be computed from climate model output, in which PM and ozone data is limited, and hence can add to our understanding on how air quality may change in the future. This has implications for understanding health related climate impacts caused by the changing climate.

Air stagnation is generally expected to become more frequent with climate change (Leung and Gustafson Jr, 2005; Horton et al., 2012, 2014; Caserini et al., 2017). Projected increases in temperature, reductions in precipitation and cyclone frequency, and shifted jet streams contribute to the projected increase in stagnation (Mickley et al., 2004; Leung and Gustafson Jr, 2005; Horton et al., 2012, 2014; Caserini et al., 2017). In addition, air stagnation and high pollution episodes tend to coincide with heatwaves during summer (Schnell and Prather, 2017), exacerbating the health impacts of each phenomenon, so understanding and trusting future changes in them is crucially important. In Horton et al. (2014), a bias correction was needed to account for climate model deficiency in simulating conditions favourable for air stagnation. Caution must be taken when analysing bias-corrected climate model output, as it cannot overcome all model errors or correct model variability (Maraun et al., 2017), and can alter relationships between variables and break conservation principles (Ehret et al., 2012). The identification of air stagnation in climate models uses meteorological variables that climate models may struggle to represent, such as precipitation and near-surface wind speeds (Flato et al., 2013). Climate model simulations are expected to better represent features of the large-scale circulation than the variables used to compute air stagnation indices. Therefore, the findings presented in this paper can be used to evaluate the reliability of air stagnation in climate models and assess their future changes. The large-scale circulation and stagnation correspondence in climate model simulations will be presented in a future article.

*Data availability.* The ERA5 reanalysis used in this study is available from the European Center for Medium-range Weather Forecasts website (https://www.ecmwf.int/en/forecasts/datasets/reanalysis-datasets/era5). The CAMS reanalysis is available from the Copernicus website (https://ads.atmosphere.copernicus.eu).

*Author contributions.* JMGP calculated the air stagnation indices as defined in Wang et al. (2016, 2018) and Huang et al. (2017). JM performed the rest of the analyses in this study and wrote the manuscript. MA, DB, RGH and CO contributed to the interpretation of the results and the writing of the paper.

*Competing interests.* The authors declare that they have no conflict of interest.

*Acknowledgements.* This work has been funded by the Spanish Ministerio de Economía, Industria y Competitividad under grant CGL2017-83198-R (STEADY) and Ministerio de Ciencia Innovación y Universidades, under grant RTI2018-096402-B-I00 (JeDiS). MA was supported

by the Program Atracción de Talento de la Comunidad de Madrid (2016-T2/AMB-1405), JMGP by a predoctoral research grant awarded by the Spanish Ministerio de Educación, Cultura y Deporte (FPU16/01972) and CO by the Ramón y Cajal Programme of the Spanish Ministerio de Economía y Competitividad (RYC-2014-15036).

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
