# Peer review of "Linking air stagnation in Europe with the synoptic- to large-scale atmospheric circulation"

_Weather and Climate Dynamics, 2021_

## Author Response (AR1)

**Authors response**

We first provide a broad overview of the main changes that address the common concerns shared by both reviewers. The main modifications to the manuscript are:

1. The introduction has been rewritten to provide a more complete analysis of the current understanding on air stagnation and its relevance for pollution levels and variability.
2. We have repeated much of the analysis using ozone ($O_3$) and PM2.5 data from the Copernicus Atmospheric Monitoring Service (CAMS) reanalysis, considering both daily and monthly mean values. Overall, the results presented for air stagnation are similar to those obtained using $O_3$ or PM2.5.
3. We have added a subsection (Section 3.3) and section (Section 5) where we compare and discuss the results obtained from pollutant data and from the air stagnation indices.
4. The abstract and conclusions have been changed to reflect the changes elsewhere in the manuscript.

(In the following the reviewers' comments appear in black with the author responses in blue)

**Reply to anonymous reviewer 1**

This manuscript examines the link between European air stagnation and the large-scale circulation and synoptic-scale features such as blocking, Rossby wave breaking, and jet locations and strength. The analysis presented appears sound, and the manuscript does present some interesting results connection these different atmospheric features. However, I am not sure of the real value of this research. It is claimed that "understanding the development of stagnant conditions is therefore crucial for studying poor air quality" and this is the justification for this study. However, I am not convinced of the importance for air quality, as multiple recent studies show there is only a weak connection between air stagnation and air pollution events, and there is also evidence that some of the other atmospheric features examined are better predictors of air pollution events (and hence more crucial).

The connections shown between stagnation and other circulation features is probably worth publishing (although I am not totally convinced), but before this manuscript is suitable for publication there needs to be a major revision of the abstract, Introduction, and conclusions that presents a more balanced view of importance of stagnation events on air pollution, and hence a more balance view of the potential impact of this study on understanding air quality and/or for the weather-climate dynamics community.

We thank the reviewer for their comments highlighting the need to better explore the air stagnation and pollution connection. We have addressed the reviewers concerns by substantially changing the introduction, abstract and conclusions, as well as repeating some of the analysis using an atmospheric composition reanalysis.

MAJOR COMMENTS

The abstract, Introduction and conclusions are full on statements that air stagnation is closely connected with air pollution, and that understanding stagnation events is key/critical for understanding air quality. However, very few of the referenced studies actually show a close connection between stagnation indices and air pollution (and many just assume there is a connection as it seems like there should be). In fact, several recent studies (that are not discussed in Introduction) actually show a weak, or at least inconsistent, relationship between air pollution and stagnation indices. The results of two of these (Kerr and Waugh (2018) and Garrido-Perez et al. (2019)) are buried in the conclusions (lines 461-462) but there are more studies: E.g., Huang et al (2018), Wang et al (2018), and Oswald et al (2015) show either weak relations between stagnation and pollution or that stagnation is not a strong predictor of pollution events. Huang et al and Wang et al are referenced but no mention is made that these studies cast doubt on importance of stagnation for air pollution events. Given multiple studies all showing weak relationship many of the claims made in the paper on importance of stagnation are not justified, and these counter studies need to be discussed in the Introduction.

In addition to limited evidence for stagnation events being major driver for pollution events, there are many recent studies showing connections of the other atmospheric features examined here (e.g jet latitude, RWB, etc) with pollution (e.g. some of references on lines 460-465). If understanding air quality is the main focus then why not relate the large-scale or synoptic-scale features to air quality data directly, rather than indirectly through stagnation events? Put another way, the importance of an atmospheric feature in explaining stagnation events is not the same as the importance of this feature on air pollution (with some features probably being more important for air pollution than for stagnation events).

This last point makes me wonder why the analysis presented was not done relating the large-scale / synoptic scale atmosphere features to surface air quality. This would directly address the issue of air quality.

From the introductory comment and major comments, we understand that referee #1 is concerned about the connection between stagnation and air pollution. He/she claims that:

(1) The abstract, Introduction and conclusions are full on statements that air stagnation is closely connected with air pollution, and that understanding stagnation events is key/critical for understanding air quality. However, several studies show a weak or

inconsistent relationship between air pollution and stagnation indices, or that stagnation is not a strong predictor of pollution events (e.g. Oswald et al. 2015; Huang et al., 2018; Kerr and Waugh, 2018; Wang et al., 2018; Garrido-Perez et al., 2019).

(2) Other atmospheric features than stagnation are better predictors of air pollution.

(3) There are many recent studies showing connections of the other atmospheric features examined here (e.g. jet latitude, RWB) with pollution. It might be more appropriate to relate the large-scale or synoptic-scale features to air quality data directly, rather than indirectly through stagnation events, in the analysis presented here.

Here we provide some comments on these points:

(1) We agree that some of the papers mentioned by the referee show weak or inconsistent relationships between air pollution and stagnation. Therefore, some discussion is needed. Here we discuss some of the findings for the US, China and Europe.

There are indeed some inconsistent results for the US, where most studies have employed Horton's air stagnation index (ASI). For instance, Oswald et al. (2015) developed statistical models to reproduce ozone exceedances at more than 80 sites in Northeastern US. Other meteorological predictors (mainly related to temperature, solar radiation and, to a lesser extent, rainfall) were more often selected than stagnation in their models. They also found correlation coefficients (R) with median values close to 0.3 for the number of stagnation events and ozone across all sites. While that value is lower than the correlations found for ozone with temperature and radiation, it exceeds that calculated for ozone and frontal passage, which is a known mechanism for the ventilation of pollution not only in the US (Leibensperger et al., 2008; Tai et al., 2010, 2012a, 2012b) but also in Europe (Ordóñez et al., 2005). On the other hand, Sun et al. (2016) found that high ozone days can occur through a subtle set of circumstances which vary among the different regions of the US, although they tend to be more favoured by the persistence of stagnation than by persistent high temperatures or the number of days since cyclone or frontal passage. Schnell and Prather (2017) showed for eastern North America in summer that stagnation and most high extremes of ozone, $PM_{2.5}$ and daily maximum temperature occur in large-scale, multiday, coherent structures. Their analyses also show that the interannual variability of air stagnation frequency closely resembles that of ozone, $PM_{2.5}$ and daily maximum temperature. However, Kerr and Waugh (2018) demonstrated that the increases in pollutant concentrations on stagnant days are small and that many pollution events are not associated with stagnation events during summer in the US. They suggested testing other indices using meteorological predictors such as temperature and boundary layer height instead of Horton's ASI.

As for the papers by Huang et al. (2018) and Wang et al. (2018) mentioned by the referee for China, the authors also started their analyses by testing Horton's index. This ASI is based on NOAA's definition of air stagnation and therefore includes a condition on the mid-tropospheric wind. Both studies reported weak relationships between air pollution in China and that index. This occurs at least partly because the strong effect of

terrain on meteorological conditions in the region leads to a decoupling between the mid and lower tropospheric layers in China. Consequently, both studies developed new stagnation indices which better represent the atmospheric conditions conducive to air pollution there. These indices use meteorological fields representing mixing or ventilation within the boundary layer, in line with the recommendations by Kerr and Waugh (2018). The association between stagnation as derived from the new indices and $PM_{2.5}$ in China was demonstrated by these studies.

This manuscript is focused on the relationships between stagnation and the large-scale to synoptic-scale circulation in Europe, where the links of pollution with both stagnation (see paragraph below) and the atmospheric circulation (see points 2 and 3) have been examined in recent analyses.

Garrido-Perez et al. (2018) found different evidence on the impact of stagnation on air pollution over most of Europe. These include winter $PM_{10}$ and summer $O_3$ enhancements of 10–16 µg m$^{-3}$ (31–58 %) and 6–10 ppb (13–23 %), respectively, during stagnant days over four out of the five regions considered here; strong build-up of both pollutants during widespread stagnation events in most regions, and a close relationship between the occurrence of stagnation and summer $O_3$ extremes (exceedances of the 95[th] percentiles) at interannual time scales, with R values from 0.54 to 0.81 for the four regions with complete time series. On the other hand, both Carro-Calvo et al. (2017) and Garrido-Perez et al. (2019) pointed to the importance of another process, namely southerly advection of polluted warm air masses, for the occurrence of ozone episodes in northern Europe under non-stagnant days. However, the occurrence of stagnation is relevant for summer ozone over most of Europe. Garrido-Perez et al. (2019) still shows that there is a connection, although with clear regional dependence, between the percentage of stagnant area in a region and summer MDA8 $O_3$ on daily time scales, with R = 0.50–0.70 in central/southern Europe and R=0.06–0.39 in the north. Garrido-Perez et al. (2018, 2019) only used Horton's index, because the other two indices considered here (i.e. those by Huang and Wang) have been created to represent the conditions leading to the accumulation of particulate matter (PM). More recently, Garrido-Perez et al. (2021) examined the relationship between the three indices and $PM_{10}$ in Europe over a somewhat longer period than the first analysis (Garrido-Perez et al., 2018). On average over 300 background stations in Europe, they found large $PM_{10}$ enhancements during stagnant days, with mean values ranging around 17.0–18.5 µg m$^{-3}$ (59–63 %) in winter and 4.0–5.5 µg m$^{-3}$ (19–28 %) in summer for these indices. The results were similar for the three indices in winter, but Horton's ASI outperformed Huang and Wang's indices during summer.

So, while we agree with the referee that the connection between stagnation and air pollutants in the region of analysis is not always strong and that other atmospheric features might be better predictors (see reply to point 2 below), there is numerous evidence that air stagnation is one of the main meteorological factors triggering high ozone and PM concentrations in Europe. Following the referee's suggestions, we have summarised some of the findings discussed here in the Introduction to provide a more balanced view of the importance of air stagnation for air quality. The abstract and conclusions have also been changed accordingly.

(2) We agree that other atmospheric features (on a variety of spatial scales) may be better predictors of air pollution. We also acknowledge some of the limitations of the indices considered here (e.g. they rely on fixed thresholds). However, this does not mean that the effect of stagnation on air pollutants in Europe is not relevant and therefore the mechanisms behind it should not be investigated. As demonstrated from the comments above, stagnation is associated with $O_3$ and PM increases and can even drive the variability of these pollutants at different time scales. Garrido-Perez et al (2018) showed strong correlations between daily $O_3$ and stagnation for most regions at daily scales (see values reported above). Such correlations are comparable to those of ozone with daily maximum temperature, which is known to be the main meteorological driver of summer maximum daily 8-h running average near-surface ozone (MDA8 $O_3$) over most regions (see Table 1 of that paper). Moreover, Garrido-Perez (2021) proved that statistical models including meteorological fields directly or indirectly represented in the three stagnation indices can reproduce 43% and 37% of the day-to-day variability of $PM_{10}$ in winter and summer, respectively. That analysis admits, however, that stagnation-related variables neglect the important effect of other meteorological variables such as temperature and humidity.

One could also compare the importance of stagnation and large-scale circulation features in triggering high levels of atmospheric pollutants. For instance, the concentration anomalies reported above for summer ozone and winter $PM_{10}$ during stagnant days are within the same order of magnitude as those found under the presence of anticyclonic systems over specific sectors (Europe or the Atlantic): 5-15 ppb for summer ozone over different regions under the presence of blocking or subtropical ridges (Ordóñez et al., 2017) and around 12 μg m$^{-3}$ for winter $PM_{10}$ over northwestern and central Europe on days with blocking (Garrido-Perez et al., 2017).

All in all, these results indicate that air stagnation is an important process to consider in Europe. The results also suggest that there might be some association between the occurrence of stagnation and high temperatures in summer, or between the occurrence of stagnation and anticyclones, indicating its potential to represent the meteorological conditions driving other meteorological extremes such as heatwaves. While ASIs have limitations, their main advantage lies in their simplicity and versatility. The main ASI considered in this study (i.e. Horton's index) can easily be calculated from meteorological fields that are available from most 3-dimensional models. In addition, the peer-review articles cited above using this index have proven that stagnation yields ozone and $PM_{10}$ increases within a similar order of magnitude as those caused by other atmospheric features in Europe.

(3) The reviewer wonders why we have not related large-scale or synoptic-scale features to air quality data directly, rather than indirectly through stagnation events.

As mentioned by the reviewer, different analyses have already related some of the circulation features discussed here to air pollution in Europe over relatively long periods. See e.g. Webber et al. (2017) for the relationship of Rossby wave breaking with $PM_{10}$ in

the UK, Garrido-Perez et al. (2017) and Ordóñez et al. (2017) for the impact of blocking and subtropical ridges on winter $PM_{10}$ and summer ozone, or Ordóñez et al. (2019) for the role of the North Atlantic jet in the variability of winter $PM_{10}$. All these analyses have shown robust relationships (although not always co-occurrence as also happens with stagnation) between pollutant concentrations and the mentioned circulation features. On the other hand, Garrido-Perez et al. (2018, 2019, 2021) have demonstrated the connection between stagnation and air pollutants, despite some limitations mentioned above.

Note that all these studies rely on the availability of air pollution data. One of the problems with most air quality data sets is the presence of trends induced by emission changes. Moreover, there is a lack of long-term time series. The studies mentioned here have used $PM_{10}$ and $O_3$ observations of up to 13 years and 18 years, respectively. The number of relatively long observational time series of $PM_{2.5}$ in Europe is even smaller than that for $PM_{10}$. As some of the relationships found between air pollution and the atmospheric circulation or stagnation seem to be robust, they could be further explored with longer air quality datasets if available from climate-chemistry and Earth system models.

Nevertheless, at this stage we still need to understand the interconnections (or lack of) among the large-scale circulation, stagnation, and air pollution. This can aid other investigations on the predictability of the atmospheric conditions driving pollution episodes or on the future evolution of such conditions in a changing climate. As shown above, there is evidence for the connections of air pollution with both the large-scale circulation and stagnation, but the connection between the large-scale and stagnation has not been established for the region of study in a systematic manner. This is what we are addressing in this manuscript. These analyses are worthwhile, because it is unclear to what extent air stagnation can characterise (or can be driven by) the large-scale and synoptic patterns that are known to influence near-surface air pollution in Europe. Furthermore, the use of a meteorological reanalysis, with several decades of data and no trends associated with emission changes (two of the main problems with most air quality datasets) is appropriate for such investigations.

According to these comments, we have substantially modified the abstract, introduction and discussion of the results. We have tried to present a balanced view of the importance of both the large/synoptic scale circulation and stagnation for air pollution. We also emphasise the need to understand the potential relationships between some of the main features of the atmospheric circulation and air stagnation in the area of study.

Finally, we have also looked into some air quality data in an effort to add further evidence to the stagnation and pollutant correspondence for Europe, as listed in detail above. We repeated some of  the composite analysis and the multilinear model for both ozone and PM2.5 variability taken from the CAMS reanalysis. The time period that the CAMS reanalysis is available is considerably shorter (around 15 years) than that for the reanalysis used to calculate the ASIs (around 40 years). Because of this, the focus

remains on the ASI results but we now have the additional comparison with pollutants, as suggested by the reviewer. For the composites, we find that the occurrence of such synoptic- to large-scale drivers drive pollution extremes (as well as stagnation increases) in many of the regions and seasons. Furthermore, we find that overall the large- to synoptic-scale predictors are able to explain a similar amount of variability in the pollutants as in the ASIs. Nonetheless, the skill of the model is somewhat lower for PM2.5 in seasons other than winter. We have added a new subsection (section 3.3) and section to the paper (section 5) in which we compare some of our results among the different ASIs and for ozone and PM2.5.

Carro-Calvo L., Ordóñez C., García-Herrera R., Schnell J.L. (2017): Spatial clustering and meteorological drivers of summer ozone in Europe. Atmospheric Environment, 167, 496-510. doi:10.1016/j.atmosenv.2017.08.050.

Oswald, E. M., L-A Dupigny-Giroux, E. M. Leibensperger, R. Poirot, J. Merrell. Climate controls on air quality in the Northeastern U.S.: An examination of summertime ozone statistics during 1993-2012. Atmospheric Environment 112, 278-288, 2015.

Tai, A. P. K., Mickley, L. J., and Jacob, D. J.: Impact of 2000–2050 climate change on fine particulate matter ($PM_{2.5}$) air quality inferred from a multi-model analysis of meteorological modes, Atmos. Chem. Phys., 12, 11329–11337, https://doi.org/10.5194/acp-12-11329-2012, 2012b.

MINOR COMMNENTS

Title:  As much of the paper focuses on blocking, ridges and Rossby-wave breaking, which are synoptic-scale and not large-scale features, the title needs to be modified.

The title has been changed to "Linking air stagnation in Europe with the synoptic- to large-scale atmospheric circulation".

Line 119. Not sure what is meant by "averaged vertically (every 75 hPa ...". Do you mean averaged vertically between 925 and 700 hPa, with data every 75 hPa?  Also, later in the sentence I think should say "zonally between 0 and 60 W", as the 0-60 is important.

This has been clarified.

Line 124: Why 20E-60W as opposed to 0-60W used for eddy-driven jet?

20E-60W was chosen as the subtropical jet tends to be identified over this region (e.g., Asiri (2020)) and less so over the North Atlantic. This has been clarified in the revised manuscript.

Asiri, M.A., Almazroui, M. & Awad, A.M. Synoptic features associated with the winter variability of the subtropical jet stream over Africa and the Middle East. *Meteorol Atmos Phys* 132, 819–831 (2020). https://doi.org/10.1007/s00703-019-00722-4

Line 165: space missing "inGarrido"

Done.

Fig 2 and 3. Can you add a box to the maps to show the region being considered?

We think that this makes the figure appear messy and may distract from the results. As the regions are not regular shapes drawing the outline is quite uneven. The regions are shown in Figure 1 for reference.

Also, I think the histogram and lines in right panels make figures hard to read. Maybe just show the curves?

Done.

Line 245-250: I could not follow this discussion, and how it related to figure 4. For example, it is stated that Rossby waves have no impact for souther regions, but the RWB and other indices have a very similar variation in all panels in fig 4.

This discussion was not directly related to Figure 4 but considering the converse situation, i.e. the change in stagnation during Rossby wave breaking events (rather than the change in RWB during stagnation shown in Figure 4). This has been made clearer in the text.

REFERENCES

Oswald E M, Dupigny-Giroux L-A, Leibensperger E M, Poirot R and Merrell J 2015 Climate controls on air quality in the northeastern US: an examination of summertime ozone statistics during 1993–2012 Atmos. Environ. 112 278–88

**Reply to anonymous reviewer 2**

GENERAL COMMENTS:

Maddison et al. examine the relationship between stagnation, defined by the Air Stagnation Index, and various dynamical features related to synoptic-scale circulation over Europe. By building a model using a MLR approach and stepwise procedure, they are able to explain a majority of the variance of stagnation frequency across different regions in Europe. They found regional heterogeneity and seasonality in the features (e.g., STJ, RWB) linked to stagnation.

I thought that the paper was well-written, and I appreciate that they tested other stagnation indices (Lines 411ff), as this has been a question of mine for quite some time. I do, however, think that the focus on stagnation is problematic and doesn't really give information about pollution events. As discussed in my comments below, I believe this paper needs a direct assessment of the link between air pollutants and the large-scale dynamical features. If these comments are heeded, I believe the paper has the potential to be relevant for forecasting pollution events with short lead times.

We thank the reviewer for their useful suggestions to improve the manuscript. We have, as the reviewer suggested, repeated a substantial part of the analysis using the CAMS atmospheric composition reanalysis and found similar results to those obtained looking at air stagnation. We address the reviewer's comments further below.

SPECIFIC COMMENTS:

• The stagnation index is problematic as it doesn't correlate well with actual pollution events [Huang et al., 2018; Kerr & Waugh, 2018; Wang et al., 2019; Garrido-Perez et al., 2019] and it's boolean, based on fairly arbitrary thresholds. The authors mention some of the known issues with the stagnation index in the Conclusions (Lines 461-462), but this comes across as a brief afterthought. In light of these issues, some of the findings contained in the study are not that groundbreaking or useful, especially for actually understanding the dynamical drivers of *pollution* events.

For example when you describe stagnation frequency and where stagnant conditions are most likely to occur (e.g., Lines 30ff, Figure 1a, Lines 211-212), I'm not surprised that stagnation frequency is lower in mid-latitude Europe (near the latitude where the jet is located and where there are mid-latitude storms) versus in the southern part of your

domain, where conditions might be more dry and where upper-level winds are not as great as in the mid-latitudes. Moreover, you also make statements like such as "air stagnation in Europe is shown to be strongly influenced by the large-scale circulation" (Line 432). Since stagnation, in essence, is just a measure of the local winds and precipitation, it doesn't seem at all surprising that local weather conditions are influenced by the large-scale circulation.

I believe that a more relevant study and one that would be impactful and important for the community would be to directly look at the relationship between pollutants and the dynamical features rather than the relationship between stagnation and the dynamical features, which may or may not have any direct relevant for surface-level pollution events. I think you could more or less repeat your methods and approach using the dynamical indices but subbing in perhaps PM and/or O3 from CAMS/ERA5. For example, you mention "stagnation is around 30% more likely to occur when there is a block or ridge present" (Lines 243-244). I would like to see a similar statement but for O3 or PM (e.g., "Extreme PM is X% more likely to occur when there is a block or ridge present"). As previously mentioned, I think this would be of wide interest to the air quality/atmospheric chemistry community.

I noticed the other reviewer pointed out a similar comment, and hopefully by addressing this comment you could kill two birds with one stone….or, more accurately, you could shut up two reviewers with one additional analysis :)

We refer to our detailed response to anonymous reviewer 1 regarding the current understanding of air stagnation and its relation to pollution events, where we argue that for Europe in particular, air stagnation is relevant for air pollution as shown in several previous studies. We have then calculated the effect of the large-scale drivers on pollution levels as suggested by the reviewer and added a section (3.3) in which we discuss them. We have also repeated the stepwise linear model for ozone and PM2.5 and added a new section to the paper ("Comparison with other ASIs and pollutants") where we discuss the results. Overall, the results agree quite well between the ASIs and pollutants, although with somewhat poorer model skill in the case of PM2.5.

References: Wang et al.
https://journals.ametsoc.org/view/journals/bams/99/1/bams-d-16-0301.1.xml

• Equations 1-3. You mention that variables such as BI_{region} and RWBI_{region} are monthly time series of blocked days. What do the distributions of these (and other) variables look like? Do they have many values = 0 and a few other values? I'm wondering if it's appropriate to use linear regression to derive the coefficients in the case that the distributions are highly skewed. Plotting some examples would easily lend insight to this question.

The distributions are not too highly skewed in the majority of cases, though 0 is the most frequent value for blocking and ridges in SW and SE. However, the model residuals are

approximately normally distributed which suggests the linear approximation is valid. The figure below shows some example distributions of the ASI, BI, RWBI and RI as well as the stepwise model residuals for certain regions and seasons (including some of the most skewed distributions).

[Figure]

TECHNICAL CORRECTIONS:

• Figure 2a,b,e - It might be helpful to include latitude labels on these maps so readers can match them with the latitudes shown in c, d, f, and g. Additionally, can you clarify the quantities being shown in the shading in these subplots? For example, you mention in the figure caption that Figure 2b shows the "ridge index (shading)" but in the text you mention "an ~20% increase in the climatological frequency of subtropical ridging over the region" (Lines 178-179). So shouldn't the quantities in shading be referred to as a departure from the index? Or am I misinterpreting?

We have added latitude labels as suggested. The shading does indeed show departures from the climatology, we have rephrased the caption to make this clearer. It now reads

"(a) Blocking index departure (shading) and composite Z500 (line contours), (b) ridge index departure (shading) and composite Z500 (line contours), and (e) Rossby wave breaking index departure (shading) and composite theta_2PVU (line contours) for stagnant days over SW. In (a), (b) and (e) the blocking, ridge and Rossby wave breaking index frequencies are presented as departures from their annual climatological frequencies, respectively".

• Figure 2a,b,e - On a related note to my above comment, the changes in climatological frequency are generally at most +/- 20% but often quite a bit less than this value. Would, for example, a 5 or 10% increase in frequency even be statistically significant? Can the authors denote statistical significance here (and elsewhere, when applicable)?

Significance has been denoted in the figures that show changes in the climatological frequencies of the blocking, wave breaking and ridge indices during stagnation (Figure 2 and 3), as well as the ASI in the presence of the large-scale drivers (Figure 8). Significances have been calculated using a Monte Carlo, bootstrapping approach (comparing the composite value obtained under the given criteria at each grid point with 5000 randomly generated composites of the same size drawn from the climatology).

• Figure 3c - I understand that the RI departure plot is based on an index that considers three separate zonal sectors. However, it looks a little strange to me to have a negative box over part of Southern Europe and then a sharp cutoff to weak positive values. I realize that this affect is an artifact of the index, but what does it imply about its suitability?

We agree that this looks strange and is an artefact of the index. However, we believe that the index is suitable for our purpose. Firstly, we tested the zonal sectors used to define the index. Once with only one region spanning the Mediterranean region, and once with two regions splitting the Mediterranean in two. The resulting composites were similar. And regarding its inclusion in the model, we use the index to identify when a ridge is present in one of the three regions on a specific day so the change from one region to the next is not present.

• Figure 6 - I think readers might benefit from having the region acronyms (e.g., SE, SW, CEU, etc.) labeled on the maps in case they forgot where these regions were.

These have been added to panel (a). Note that this is now Figure 7 in the revised version of the manuscript.

---

## Author Response (AR2)

The authors thank the reviewer for their comments on the revised manuscript. In the following the reviewers' comments appear in black with the authors' responses in blue.

In this paper, the authors examine the connections among air stagnation, PM2.5 and O3 concentrations, and various dynamical indices related to the synoptic scale circulation over Europe. They demonstrate that the synoptic/large-scale circulation can explain >50% of the variability of monthly variability of stagnation and pollutant concentrations. Overall, the manuscript is well-written, and I appreciate the work the authors have already done responsive to the reviewers' requests from the discussion stage. I have a handful of comments and questions, mainly about the MLR methodology and the robustness of results to different timescales.

Major comments:
• I hypothesize that several of the dynamical indices examined herein are correlated. While you address multicollinearity using VIFs (Lines 184-187), have you tested the sensitivity of your selected predictors (Figure 7) by using a more conservative VIF under 5? Could restricting the VIF threshold create a more parsimonious model?

We have tested the sensitivity of the model using different VIFs. The models are very similar in each case. Using a VIF of 2 (the most restrictive value we tested), and considering the leading six predictors selected by the model in each region and season, a total of only 11 out of the 120 predictors (considering all regions and seasons) in the European model with a VIF of 5 are removed for being too highly correlated with other predictors. This will not have a large impact on the model as used for our purpose. Indeed, the model using a VIF of 2 has skill (R2) at most 0.03 less than the model using a VIF of 5. To highlight this in the article, we have added the sentence "Using a more restrictive threshold of 2.0 for the VIF does not change the conclusions drawn from the results presented in this article." (Lines 186-187).

• Have you accounted for long-term secular trends in the data in your MLR approach? I expect that variables related to the ASI and those related to your dynamical various indices have changed over the ~40 year period and these changes might impact your results.

We have not accounted for trends in the MLR directly by removing trends in the various variables or anything similar. As there is no single way to detrend, and given that we would need to decide how to detrend each variable, we chose not to as this could add additional uncertainty to the model interpretation. Furthermore, detrending may not be particularly useful here, because long-term trends in the dynamical fields (block/RWB/ridges/etc) are overall small, or at least smaller than internal variations (or than interannual variability).

We have tested how the model performs during different time periods within the reanalysis period (using fewer years to fit the model) and the skill does not show any large change for the different periods. This can also be seen by comparing the model skill for the Horton ASI across the whole period (Figure 6) with the skill when calculating using only 16 years of data (Figure 9) where no large difference is observed.

• Figure 5 and corresponding discussion: Are the "extreme ozone days" and "extreme PM2.5 days" defined for each season or using all seasons? I am specifically trying to make sense of the very small impact of BI and RI on ozone for winter and wondering if this finding is simply an artifact of ozone's pronounced seasonal cycle.

The extreme pollutant days are defined for each season separately to take into account the seasonal cycle of the pollutants. We have made this clear in the text (Lines 281-282). We would not expect blocks to be related to high ozone days in winter as weather conditions during such events favor ozone reduction (Ordóñez et al. 2017). On the other hand, subtropical ridges have been shown to have a moderate impact on ozone concentrations in winter (Ordóñez et al. 2017) in agreement with the presented results.

• Given that the EU's air quality directives are based on a mixture of hourly/daily (e.g., O3) and annual thresholds, have you considered the impact of the dynamical features of interest on *daily* pollutant concentrations rather than the monthly mean values you currently use (Lines 483-484)? If your results and overall conclusions not robust to the choice of daily versus monthly data, does this imply something about what controls PM2.5 and O3 (and stagnation) variability on daily versus longer timescales?

The stepwise model results cannot be compared to daily data as we are considering the monthly variability of the pollutants (and dynamical indices, many of which are binary fields on daily scales). However, we consider the impact of blocks and ridges on daily pollutant concentrations in section 3.3 (and Figure 5), where we show that there is a clear influence of both blocks and ridges on pollutant concentrations. This supports that the signals we report on monthly mean concentrations stem from daily anomalies in the pollutants linked to atmospheric circulation in a similar fashion as on monthly scales. Therefore, monthly values largely result from the recurrence of daily anomalies in the month, which in turn depends on the monthly frequency of occurrence of those dynamical features.

Minor comments
• Line 180: typo "skilful"
Changed
• Figure 4: letters (a)-(e) are more difficult to read than on the rest of your figures
We have made the text larger here.
• You state that the ridge index is defined south of 50˚N (55˚N in summer); this is shown in Figure 2b. Why are there values for ASI defined north of 50˚N in the RI plots of Figure 8a, d?
The field shown is the air stagnation index, composited on days defined as having ridges, blocking, or Rossby wave breaking, and hence spans all latitudes. This has been made clearer in the text (added "and are shown in Figure 8" at Line 453).

References:

C. Ordóñez et al.: Regional responses of surface ozone in Europe to the location of high-latitude blocks and subtropical ridges. Atmos. Chem. Phys., 17, 3111–3131, 2017